# Dynamics of genomic innovation in the unicellular ancestry of animals

Xavier Grau-Bové[1,2]*, Guifré Torruella[3], Stuart Donachie[4,5], Hiroshi Suga[6], Guy Leonard[7], Thomas A Richards[7], Iñaki Ruiz-Trillo[1,2,8]*

[1]Institut de Biologia Evolutiva (CSIC-Universitat Pompeu Fabra), Barcelona, Catalonia, Spain; [2]Departament de Genètica, Microbiologia i Estadística, Universitat de Barelona, Barcelona, Catalonia, Spain; [3]Unité d'Ecologie, Systématique et Evolution, Université Paris-Sud/Paris-Saclay, AgroParisTech, Orsay, France; [4]Department of Microbiology, University of Hawai'i at Mānoa, Honolulu, United States; [5]Advanced Studies in Genomics, Proteomics and Bioinformatics, University of Hawai'i at Mānoa, Honolulu, United States; [6]Faculty of Life and Environmental Sciences, Prefectural University of Hiroshima, Hiroshima, Japan; [7]Department of Biosciences, University of Exeter, Exeter, United Kingdom; [8]ICREA, Passeig Lluís Companys, Barcelona, Catalonia, Spain

*For correspondence: xavier. graubove@gmail.com (XG-B); inaki.ruiz@multicellgenome.org (IR-T)

Competing interests: The authors declare that no competing interests exist.

**Abstract** Which genomic innovations underpinned the origin of multicellular animals is still an open debate. Here, we investigate this question by reconstructing the genome architecture and gene family diversity of ancestral premetazoans, aiming to date the emergence of animal-like traits. Our comparative analysis involves genomes from animals and their closest unicellular relatives (the Holozoa), including four new genomes: three Ichthyosporea and *Corallochytrium limacisporum*. Here, we show that the earliest animals were shaped by dynamic changes in genome architecture before the emergence of multicellularity: an early burst of gene diversity in the ancestor of Holozoa, enriched in transcription factors and cell adhesion machinery, was followed by multiple and differently-timed episodes of synteny disruption, intron gain and genome expansions. Thus, the foundations of animal genome architecture were laid before the origin of complex multicellularity – highlighting the necessity of a unicellular perspective to understand early animal evolution.

## Introduction

The transition from a unicellular organism to the first multicellular animal, more than 600 million years ago (*Budd and Jensen, 2017*; *dos Reis et al., 2015*), marks one of the most radical evolutionary innovations within the eukaryotes. Although multicellularity has independently evolved multiple times in the eukaryotic lineage, the highest levels of organismal complexity, body plan diversity and developmental regulation are found in the Metazoa (*Grosberg and Strathmann, 2007*). Key advances in the study of animal origins have been made by comparing the genomes of early branching metazoa, such as cnidarians, ctenophores or sponges (*Putnam et al., 2007*; *Srivastava et al., 2010a*; *Moroz et al., 2014*; *Srivastava et al., 2008*; *Fortunato et al., 2014*), with their closest unicellular relatives in the Holozoa clade, such as the choanoflagellates *Monosiga brevicollis* and *Salpingoeca rosetta* (*King et al., 2008*; *Fairclough et al., 2013*), and the filasterean *Capsaspora owczarzaki* (*Suga et al., 2013*) (*Figure 1*). By focusing on the transition, it is possible to determine which genomic innovations occurred at the origin of metazoa, and whether it required the invention of novel genes or structural features.

**eLife digest** Hundreds of millions of years ago, some single-celled organisms gained the ability to work together and form multicellular organisms. This transition was a major step in evolution and took place at separate times in several parts of the tree of life, including in animals, plants, fungi and algae.

Animals are some of the most complex organisms on Earth. Their single-celled ancestors were also quite genetically complex themselves and their genomes (the complete set of the organism's DNA) already contained many genes that now coordinate the activity of the cells in a multicellular organism.

The genome of an animal typically has certain features: it is large, diverse and contains many segments (called introns) that are not genes. By seeing if the single-celled relatives of animals share these traits, it is possible to learn more about when specific genetic features first evolved, and whether they are linked to the origin of animals.

Now, Grau-Bové et al. have studied the genomes of several of the animal kingdom's closest single-celled relatives using a technique called whole genome sequencing. This revealed that there was a period of rapid genetic change in the single-celled ancestors of animals during which their genes became much more diverse. Another 'explosion' of diversity happened after animals had evolved. Furthermore, the overall amount of the genomic content inside cells and the number of introns found in the genome rapidly increased in separate, independent events in both animals and their single-celled ancestors.

Future research is needed to investigate whether other multicellular life forms – such as plants, fungi and algae – originated in the same way as animal life. Understanding how the genetic material of animals evolved also helps us to understand the genetic structures that affect our health. For example, genes that coordinate the behavior of cells (and so are important for multicellular organisms) also play a role in cancer, where cells break free of this regulation to divide uncontrollably.

We now know that the animal ancestor was already a genomically complex organism, with a rich complement of genes encoding proteins related to a multicellular lifestyle. These include transcription factors, extracellular matrix components and intricate signaling pathways that were previously considered animal-specific, but were already poised to be co-opted for multicellularity when animals emerged (*Fairclough et al., 2013*; *Suga et al., 2013*; *Richter and King, 2013*; *Manning et al., 2008*; *Suga et al., 2012*; *de Mendoza et al., 2013*; *Sebé-Pedrós et al., 2017*). Suggestively, detailed analyses of the transcriptomic and proteomic regulatory dynamics of *Capsaspora* and *Salpingoeca* showed that these genes are frequently implicated in the transition to life stages reminiscent of multicellularity – aggregative in *Capsaspora* (*Sebé-Pedrós et al., 2013*, *Sebé-Pedrós et al., 2016a*), and clonal colonies in *Salpingoeca* (*Fairclough et al., 2013*). Furthermore, the genome architectures of extant Metazoa are, in many aspects, markedly different from most other eukaryotes: they have larger genomes (*Elliott and Gregory, 2015a*), containing more (*Csuros et al., 2011*) and longer introns (*Elliott and Gregory, 2015a*) that can sustain alternative splicing-rich transcriptomes (*McGuire et al., 2008*; *Irimia and Roy, 2014*), have richer complements of repetitive sequences such as transposable elements (*Elliott and Gregory, 2015b*) and are structured in ancient patterns of gene linkage associated with transcriptional co-regulation (*Irimia et al., 2012*; *Simakov et al., 2013*) – e.g., the Homeobox clusters (*Ferrier, 2016*). The relationship between these patterns of genome evolution and multicellularity is, however, unclear: these traits are not exclusive of animals (*cf.* (*Curtis et al., 2012*; *Shoguchi et al., 2013*; *Michael, 2014*; *de Mendoza et al., 2015*; *French-Italian Public Consortium for Grapevine Genome Characterization et al., 2007*)); the existence of secondarily reduced genomes in animals (smaller, gene-compact, less repetitive) in animals blurs their link with organismal complexity (*Simakov and Kawashima, 2017*; *Seo et al., 2001*; *Petrov et al., 1996*); and non-adaptive scenarios can explain the emergence of genomic complexities as a consequence drift-enhancing population-genetic environments (*Lynch and Conery, 2003*; *Lynch, 2002*, *Lynch, 2007*). Establishing the timeline of genome architecture evolution in the

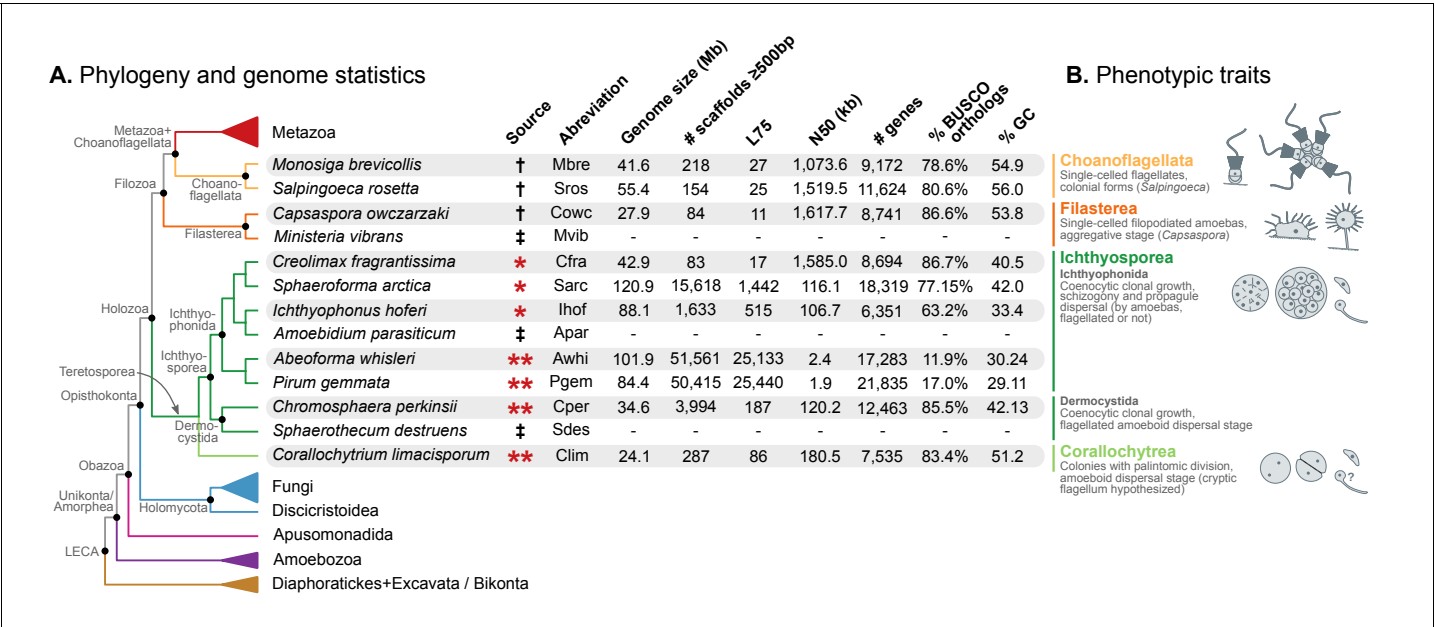

**Figure 1.** Evolutionary framework and genome statistics of the study. (**A**) Schematic phylogenetic tree of eukaryotes, with a focus on the Holozoa. The adjacent table summarizes genome assembly/annotation statistics. Data sources: red asterisks denote Teretosporea genomes reported here; double asterisks denote organisms sequenced for this study; † previously sequenced genomes (*King et al., 2008*; *Fairclough et al., 2013*; *Suga et al., 2013*); ‡ organisms for which transcriptomic data exists but no genome is available (*Torruella et al., 2015*). (**B**) Overview of the phenotypic traits of each group of unicellular Holozoa, focusing on their multicellular-like characteristics. For further details, see (*Torruella et al., 2015*; *Mendoza et al., 2002*; *Marshall et al., 2008*; *Glockling et al., 2013*). *Figure 1—source data 1* and *2*.

The following source data and figure supplement are available for figure 1:

**Source data 1.** Table of genome structure statistics, from the data-set of eukaryotic genomes used in the study.

**Source data 2.** List of genome and transcriptome assemblies and annotations, including abbreviations, taxonomic classification and data sources.

**Figure supplement 1.** Comparisons of gene length of one-to-one orthologs from pair-wise comparisons of all 10 unicellular Holozoa.

ancestry of Metazoa is thus essential to understand to which extent genomic complexity is linked to multicellularity.

Overall, gene content has been extensively studied in the unicellular ancestry of animals, but less attention has been devoted to the evolution of genome architecture in this period – covering features such as the repetitive content, intron creation and synteny conservation (although *cf.* (*King et al., 2008*; *Irimia et al., 2012*)). This bias is partly due to the multi-million year gap separating animals from their unicellular relatives and the limited genome sampling of unicellular holozoans. We now know several examples of the effects of such limitations. For instance, our view of the transcription factor repertoire of the animal ancestor was confounded by the gene losses of *Monosiga*, which only became evident when *Capsaspora* genome was analyzed (*Sebé-Pedrós et al., 2011*); and the same happened with the ancestral animal diversity of cadherin and integrin adhesion systems before genomes from choanoflagellates and *Capsaspora* were analyzed (*Nichols et al., 2012*; *Sebé-Pedrós et al., 2010*). Therefore, comparative genomics studies are highly sensitive to taxonomic biases, meaning that rare genomic changes can remain elusive, and more frequent events can manifest saturated evolutionary signals. To overcome these limitations, we analyze the genomes of the third lineage of close unicellular relatives of animals, the Teretosporea, composed of Ichthyosporea and *Corallochytrium limacisporum* (*Torruella et al., 2015*).

As the earliest-branching holozoan clade, Teretosporea are in a key phylogenetic position to complement our current view of premetazoan evolution. Interestingly, they display a developmental mode that radically differs from choanoflagellates and filastereans: many ichthyosporeans have a

multinucleate coenocytic stage (*Mendoza et al., 2002*; *Marshall et al., 2008*), and *Corallochytrium* develops colonies by binary, palintomic, cell division (*Raghukumar, 1987*). In both cases, completion of the life cycle frequently involves release of propagules that restart the clonal proliferation (*Mendoza et al., 2002*; *Marshall et al., 2008*). In addition, the ichthyosporean *Creolimax fragrantissima* exhibits many features reminiscent of animals, such as transcriptional regulation of cell type differentiation or synchronized nuclei division during its development (*de Mendoza et al., 2015*; *Suga and Ruiz-Trillo, 2013*).

Here, we present the complete genomes of four newly sequenced organisms: *Corallochytrium limacisporum* and the ichthyosporeans *Chromosphaera perkinsii* (gen. nov., sp. nov.), *Pirum gemmata* and *Abeoforma whisleri*. These are added to the already available *Creolimax fragrantissima*, *Ichthyophonus hoferi* and *Sphaeroforma arctica* (*de Mendoza et al., 2015*; *Torruella et al., 2015*) (Ichthyosporea), and to the afore-mentioned *Salpingoeca rosetta*, *Monosiga brevicollis* (choanoflagellates) and *Capsaspora owczarzaki* (Filasterea), totaling 10 unicellular holozoan genomes (*Figure 1*).

Our aim is to provide new insights into the evolutionary dynamics of the genome in the ancestral unicellular lineage leading to animals, at two broad levels: gene family origin and diversification, and conservation of genome architectural features. We address the origin of the large and intron-rich animal genomes, changes in gene linkage (microsynteny), and ancient patterns of gene family diversification. The leitmotiv of these analyses is to identify and date genomic novelties along the ancestry of Metazoa, aiming to understand the foundations of the transition to multicellularity. The emerging picture from this comparative study is one of punctuated, differently-timed bursts of innovation in genome content and structure, occurring in the unicellular ancestry of animals.

## Results

### Four new genomes of unicellular relatives of animals

We obtained the complete nuclear genome sequences of *Corallochytrium limacisporum* and the ichthyosporeans *Chromosphaera perkinsii*, *Pirum gemmata* and *Abeoforma whisleri*. For all these taxa, we sequenced genomic DNA from axenic cultures using Illumina paired-end and mate-pair reads, which were assembled using Spades (*Nurk et al., 2013*). Gene annotation was performed using a combination of de novo gene predictions and transcriptomic evidence derived from RNA sequencing experiments (see Methods). Of the four genomes presented here, *Corallochytrium* (24.1 Mb) and *Chromosphaera* (34.6 Mb) have the highest completeness and contiguity (*Figure 1*). Specifically, *Corallochytrium* has 7535 genes and 83.4% of the BUSCO paneukaryotic gene set (a proxy to genome completeness (*Simão et al., 2015*)), and 75% of the assembly length is covered by 86 scaffolds (L75 statistic). *Chromosphaera* has 12,463 annotated genes comprising 85.5% of the BUSCO set, and its L75 statistic is 187 scaffolds. In contrast, *Abeoforma* and *Pirum* have larger genome assemblies (101.9 and 84.4 Mb), but these are fragmented (L75 = 25,133 and 25,440 scaffolds) and incomplete (11.9% and 17.0% of BUSCO). These lower contiguities are reflected in their partial gene predictions (*Figure 1A*, *Figure 1—figure supplement 1*), which consequently hindered the detection of BUSCO orthologs.

Overall, together with *Capsaspora*, the two choanoflagellates and three already available ichthyosporeans, our expanded dataset now comprises 10 genomes from all unicellular Holozoa lineages – eight more than in previous genome analyses (*Fairclough et al., 2013*; *Suga et al., 2013*).

### The new *Chromosphaera* (gen. nov.) helps resolve the phylogeny of Holozoa

To have a robust phylogenetic framework for our comparative analyses, we investigated the phylogenetic relationships between holozoans with a phylogenomic analysis based on the dataset developed in *Torruella et al. (2015)*. We classified the newly identified *Chromosphaera perkinsii* (gen. nov., sp. nov.) as a member of Ichthyosporea, in the order Dermocystida, as it clusters with *Sphaerothecum destruens* in our phylogenomic analysis (*Figure 2*; BS = 100%, BPP = 1). Therefore, *Chromosphaera*, isolated from shallow marine sediments in Hawai'i, is the first described putatively freeliving dermocystid Ichthyosporea. Indeed, all described dermocystids are strict vertebrate parasites, whereas ichthyophonids are typical animal commensals or parasites (although free-living species

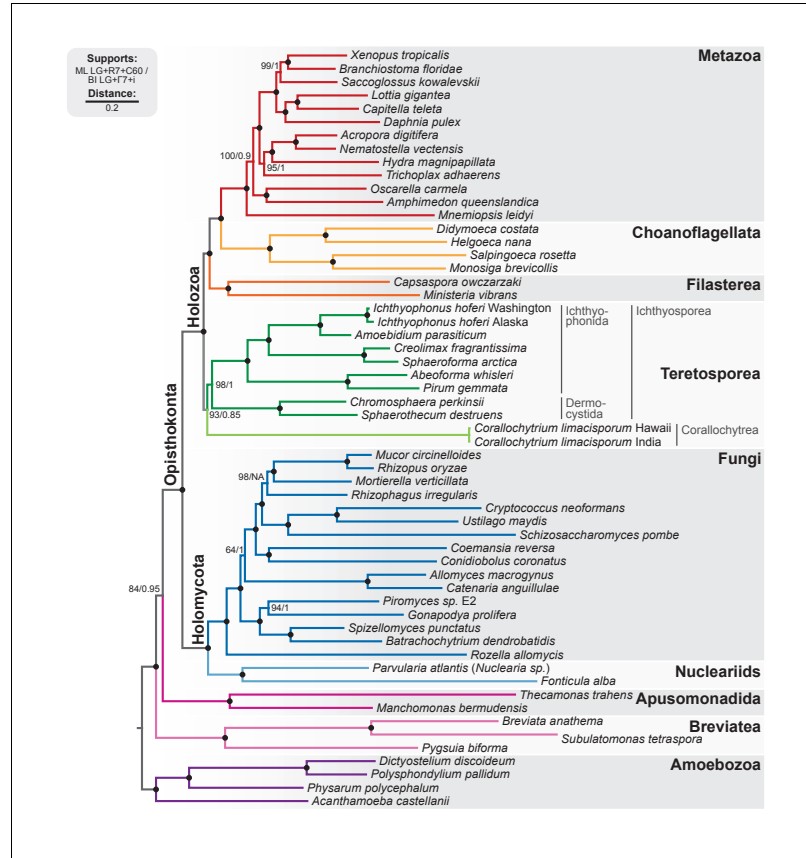

**Figure 2.** Phylogenomic tree of Unikonta/Amorphea. Phylogenomic analysis of the BVD57 taxa matrix. Tree topology is the consensus of two Markov chain Monte Carlo chains run for 1231 generations, saving every 20 trees and after a burn-in of 32%. Statistical supports are indicated at each node: (i) non-parametric maximum likelihood ultrafast-bootstrap (UFBS) values obtained from 1000 replicates using IQ-TREE and the LG + R7+C60 model; (ii) Bayesian posterior probabilities (BPP) under the LG+Γ7 + CAT model as implemented in Phylobayes. Nodes with maximum support values (BPP = 1 and UFBS = 100) are indicated by a black bullet. See *Figure 2—figure supplement 1* for raw trees with complete statistical supports. *Figure 2—source data 1*.

The following source data and figure supplement are available for figure 2:

**Source data 1.** BVD57 phylogenomic dataset (*Torruella et al., 2015*) including 87 unaligned protein domains (with PFAM accession number) per species.

**Figure supplement 1.** Phylogenomic analysis of the BVD57 matrix using (**A**) IQ-TREE maximum likelihood and the LG + R7+C60 model (supports are SH-like approximate likelihood ratio test/UFBS, respectively); (**B**) IQ-TREE maximum likelihood and the LG + R7+PMSF model (fast CAT approximation; non-parametric bootstrap supports); and (**C**) Phylobayes Bayesian inference under the LG+Γ7 + CAT model (BPP supports).

have been described and some have been identified in environmental surveys of marine microbial eukaryotic diversity) (*del Campo and Ruiz-Trillo, 2013*; *Glockling et al., 2013*).

Our analysis confirms our previous results with regards to the phylogenetic relationships within Holozoa: the Teretosporea, comprising Ichthyosporea and the small free-living osmotroph *Corallochytrium* (*Raghukumar, 1987*), are a sister-group to all the other holozoans (filastereans, choanoflagellates and animals) with improved statistical support (*Figure 2*; BS = 93%, BPP = 0.85). The monophyly of Teretosporea rejects alternative scenarios such as the 'Filasporea' hypothesis (a grouping of Filasterea + Ichthyosporea) (*Ruiz-Trillo et al., 2008*; *Liu et al., 2009*) or the status of *Corallochytrium* as an independent opisthokont lineage.

## Trends in the evolution of genome size, synteny and gene conservation across Holozoa

### Independent increases in genome size in Metazoa and unicellular holozoans

We found that Metazoa typically have larger genomes than their unicellular relatives: early-branching animals are within the 300–500 Mb range (*Elliott and Gregory, 2015a*; *Simakov and Kawashima, 2017*) and most unicellular holozoans have relatively compact genomes, like *Corallochytrium*, *Capsaspora* or *Chromosphaera* (24.1, 27.9 and 34.6 Mb, respectively; *Figure 3A*). There are, however, a few exceptions in the Ichthyosporea: *Sphaeroforma*, *Abeoforma*, *Pirum* and *Ichthyophonus* have genomes in the 84.4–120.9 Mb range (using assembly length as a proxy to genome size), sometimes larger than some secondarily simplified early-branching animals like *Trichoplax adhaerens* (~100 Mb) or *Oscarella carmela* (57 Mb; *Figure 3A*) (*Srivastava et al., 2008*; *Simakov and Kawashima, 2017*).

A parsimonious scenario for genome size evolution would imply an holozoan ancestor with a fairly compact genome, in line with the values of *Corallochytrium*, *Capsaspora* and *Chromosphaera* (24.1–34.6 Mb), followed by secondary genome expansions in ichthyosporeans (the stem lineage of ichthyophonids, and then again in individual species) and possibly *Salpingoeca* (55.4 Mb). The largest unicellular holozoan assembled genomes fall short of the inferred C-values of ancestral Metazoa (~300 Mb) (*Simakov and Kawashima, 2017*), thus indicating another genome expansion at the origin of multicellularity.

Transposable element (TE) invasions partially explain the inflations in genome size and can carry the signal of the independent expansions (*Elliott and Gregory, 2015b*). Indeed, 5–9% of the genome of *Salpingoeca*, *Sphaeroforma*, *Abeoforma* and *Pirum* are covered by TEs, whereas other holozoans are below 2.5% (*Figure 3A*). Unicellular holozoan have diverse TE complements, ranging between 42 families in *Corallochytrium* to >400 in *Pirum* or *Abeoforma* (*Figure 3—figure supplement 1*; [*Carr and Suga, 2014*]); and ~31% of these families are shared with metazoan genomes (*Figure 3—figure supplement 2*). In *Salpingoeca*, *Pirum* and *Abeoforma*, we found species-specific small sets of TE families, sharing high sequence identity, that accounted for the vast majority of copies (*Figure 3B*). This signaled recent TE invasions, and, therefore, independent contributions to genome expansion. There were hints of older TE propagation events in *Sphaeroforma* and *Pirum*, with a long tail of low-similarity TE copies (*Figure 3B*). In *Abeoforma* and *Pirum*, TEs and other simple repeats comprised up to 17% and 34% of the genome, accompanied by unusually AT-biased nucleotide compositions (*Figure 1A*). However, the exact repetitive fraction of *Abeoforma* and *Pirum* genomes cannot be exactly quantified: their highly repetitive nature has contributed to their fragmented and incomplete assemblies (*Figure 1A*, *Figure 1—figure supplement 1*) (*Treangen and Salzberg, 2011*), which hinders the annotation of TEs and simple repeats. Finally, the smaller genomes of *Corallochytrium* and *Chromosphaera* were largely depleted of repetitive/satellite regions and TEs (1.8% and 3.8% of their genomes). This finding, together with their reduced intron content (see below, *Figure 4*) suggests a secondary streamlining process.

### Synteny conservation across holozoan lineages is rare, except in *Capsaspora*

Ancestral conservation of gene linkage at the local level (microsynteny) is common in Metazoa, frequently due to coordinated *cis*-regulation (*Irimia et al., 2012*; *Simakov et al., 2013*). Following this reasoning, we analyzed the microsyntenic gene pairs of unicellular holozoan genomes (*Figure 3C*), expecting higher degrees of conservation within lineages than across them. This hypothesis held true for the *Salpingoeca-Monosiga* genome pair, but we found little or no conservation in almost all inter-specific comparisons of ichthyosporeans and *Corallochytrium*. There were, however, two exceptions: *Creolimax-Sphaeroforma* (sibling species; 907 syntenic orthologous genes) and, to a lesser extent, *Chromosphaera-Corallochytrium* (72 genes). In the case of the closely-related *Pirum* and *Abeoforma*, their fragmented genomes hindered the gene order analyses and yielded low synteny conservation values.

In contrast, the analysis of microsynteny in *Capsaspora* revealed remarkable across-lineage conservation with the distant teretosporeans *Chromosphaera* and *Corallochytrium* (142 and 129 genes, respectively). Moreover, and to a lesser degree, *Capsaspora* also retains a few shared linked gene pairs with *Trichoplax*, the cnidarians *Aiptasia* sp., *Nematostella vectens*, and the sponges *Amphimedon queenslandica* and *Oscarella carmela* (*Figure 3C*, *Figure 3—figure supplement 3*). A

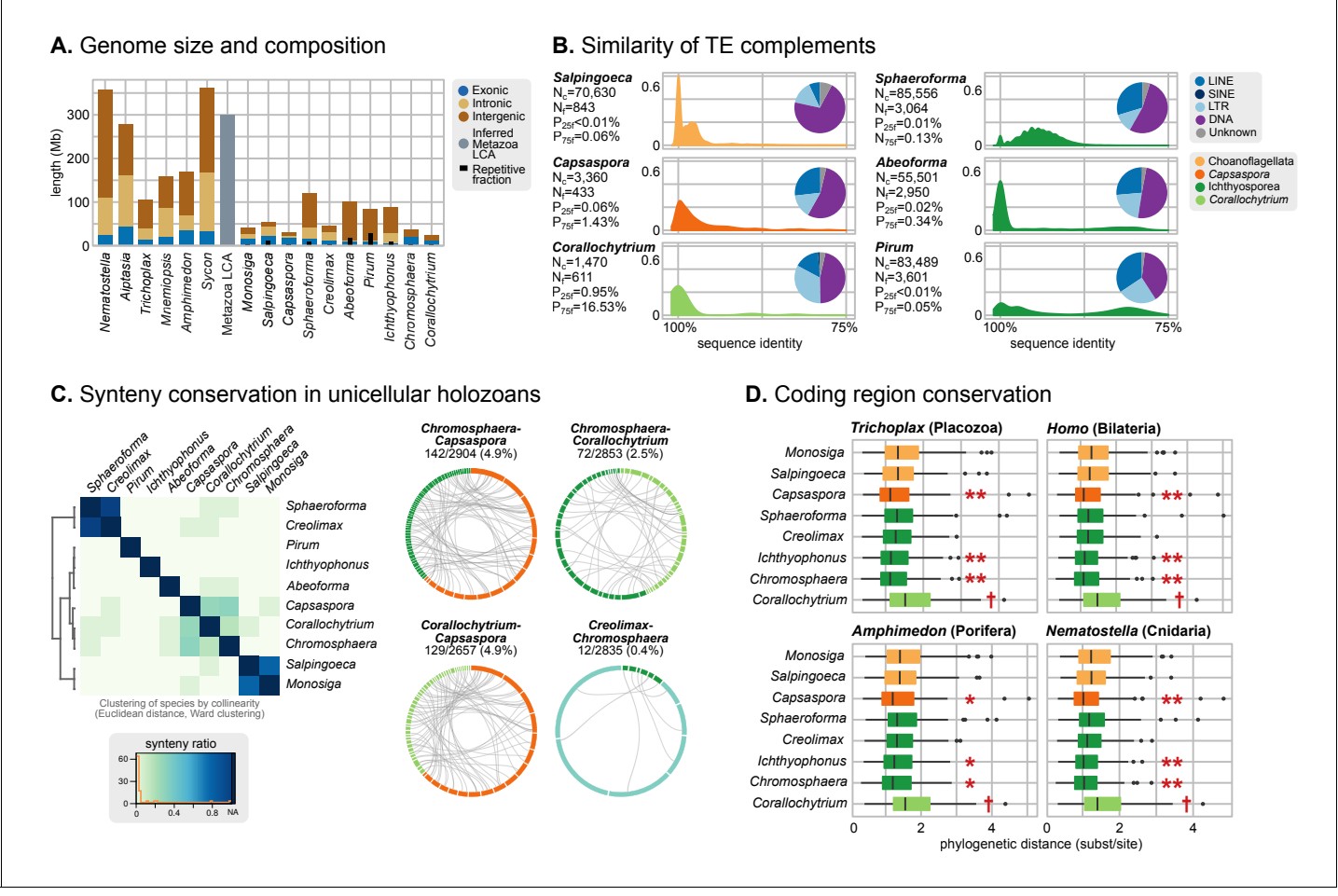

**Figure 3.** Patterns of genome evolution across unicellular Holozoa. (**A**) Genome size and composition in terms of coding exonic, intronic and intergenic sequences of unicellular holozoan and selected metazoans. Percentage of repetitive sequences shown as black bars. Genome size of the Metazoa LCA (gray bar) from (**Simakov and Kawashima, 2017**) (exonic, intronic and intergenic composition not known). (**B**) Profile of TE composition for selected organisms. Density plots indicate the sequence similarity profile of the TE complement in each organism. Embedded pie-charts denote the relative abundance, in nucleotides, of the main TE superclasses in each genome: retrotransposons (SINE, LINE and LTR), DNA transposons (DNA) and unknown. $N_c$: total number TE copies in the genome; $N_f$: number of families to which these belong; $P_{25f}$ and $P_{75f}$: percentage of most-frequent TE families that account for 25% and 75% of the total number of TE copies, respectively. (**C**) Heatmap of pairwise microsynteny conservation between 10 unicellular holozoan genomes. Species ordered according the number of shared syntenic genes (Euclidean distances, Ward clustering). At the right: selected pairwise comparisons of syntenic single-copy orthologs between unicellular holozoan genomes. Numbers denote number of syntenic genes, total number of single-copy orthologs, and proportions (%) of syntenic genes per the compared orthologs. Circle segments are scaffolds sharing ortholog pairs, connected by gray lines. (**D**) Phylogenetic distances between unicellular holozoans and four selected animals: *Homo sapiens*, *Nematostella vectensis*, *Trichoplax adhaerens* and *Amphimedon queenslandica*. Red asterisks denote organisms that have lower phylogenetic distances to metazoans than one (single asterisk) or both choanoflagellates (double asterisks) (*p* value < 0.05 in Wilcoxon rank sum test). † indicates significantly higher distances between *Corallochytrium* and metazoans. *Figure 1—source data 1*, *Figure 3—source data 1*, *2* and *3*.

The following source data and figure supplements are available for figure 3:

**Source data 1.** Annotated repetitive sequences from 10 unicellular Holozoa genomes.

**Source data 2.** List of annotated transposable element families in 10 unicellular Holozoa genomes, with copy counts.

**Source data 3.** List of annotated transposable element families shared between the genomes of 10 unicellular holozoans and 11 animals, including the number of species where the TE family is present.

**Figure supplement 1.** Profile of TE composition of unicellular Holozoa.

**Figure supplement 2.** Shared TEs between unicellular Holozoa and animal genomes.

*Figure 3 continued on next page*

*Figure 3 continued*

**Figure supplement 3.** Heatmap of pairwise ratios of ortholog collinearity between 10 unicellular holozoan genomes.

notable example of ancestral microsynteny is that of integrins: heterodimeric transmembrane proteins involved in cell-to-matrix adhesion and signaling in animals that are also present in unicellular Holozoa (*de Mendoza et al., 2015*; *Sebé-Pedrós et al., 2010*). Indeed, integrin-α and integrin-β genes from *Corallochytrium* (one pair) and *Capsaspora* (four pairs) are in a conserved head-to-head arrangement of likely holozoan origin. Incidentally, *Capsaspora*'s pairs of collinear α/β integrins co-express during its life cycle (*Sebé-Pedrós et al., 2013*), a typical cause of microsynteny conservation in animals (*Irimia et al., 2012*). Overall, gene linkage of most extant holozoans appears to be markedly different from their common ancestor, with specific gene pairings arising in Metazoa (*Irimia et al., 2012*; *Simakov et al., 2013*), choanoflagellates and some ichthyophonids. In contrast, *Capsaspora* harbors a relatively slow-evolving genome in terms of synteny conservation.

## Coding sequence conservation patterns vary across holozoan lineages

Finally, we examined the level of coding sequence conservation between unicellular holozoans and animals. We aimed to contrast the patterns of conservation at the structural level (outlined above) with those of the genic regions. Using 143 phylogenies of paneukaryotic orthologous genes, we examined the pairwise distances between unicellular holozoans and *Homo sapiens* (bilaterian), *Amphimedon* (sponge), *Nematostella* (sea anemone) and *Trichoplax* (placozoan) (*Figure 3D*). In all comparisons, *Capsaspora*, *Chromosphaera* and *Ichthyophonus* accumulated fewer amino-acidic substitutions per alignment position than choanoflagellates since their divergence from animals (p<0.05 in Wilcoxon rank sum test). Conversely, *Corallochytrium* was singled out as the taxon with more cumulative amino acid differences with animals. Thus, the analysis of coding sequence conservation across holozoans—a genomic trait fundamentally unrelated to synteny—also attests to *Capsaspora*'s slower pace of genome change.

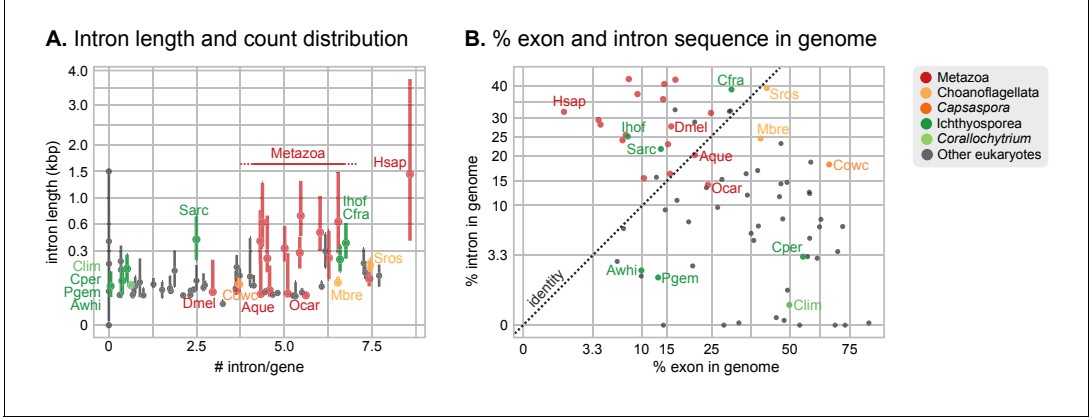

**Figure 4.** Intron abundance in eukaryotes. (**A**) Distribution of intron lengths and number of introns per gene in selected eukaryote genomes. Dots represent median intron lengths and vertical lines delimit the first and third quartiles. Color code denotes taxonomic assignment. Species abbreviation as in *Figure 1* and *Figure 2—source data 1*. (**B**) Fraction of the genome covered by introns and exons in selected eukaryotes. Dotted line represents the identity between both values. Color code denotes taxonomic assignment. *Figure 1—source data 1*.

# Intron evolution in Holozoa: two independent 'great intronization events'

## Evolution of intron structure

Intron-rich genomes are a hallmark of Metazoa. Indeed, the last common ancestor (LCA) of Metazoa is inferred to have had the highest intron density among eukaryotes, due to a process of continuous intron gain starting in the last eukaryotic common ancestor (LECA) (*Csuros et al., 2011*; *Carmel et al., 2007*). The high intron density of multicellular animals has been linked to their higher organismal complexity, as it enables frequent alternative splicing (AS) and richer transcriptomes (*Rogozin et al., 2012*; *Barbosa-Morais et al., 2012*; *Irimia et al., 2009*; *Nilsen and Graveley, 2010*), provides physical space for transcription regulatory sites (*Le Hir et al., 2003*; *Sebé-Pedrós et al., 2016b*), and facilitates the diversification of gene families by exon shuffling (*Liu et al., 2005*). The dominance of weak splice sites inferred at the intron-rich ancestral Metazoa reinforces the proposed role of alternative splicing as an important source of transcriptomic innovation at the dawn of animal multicellularity (*Csuros et al., 2011*; *Irimia et al., 2007*).

Our expanded set of unicellular holozoan genomes provides an ideal framework to investigate the emergence of the high intron densities found in animal genomes. Our survey of intron richness across eukaryotes identifies a high number of introns per gene in many ichthyosporeans, choanoflagellates and animals (*Figure 4A*). Moreover, *Creolimax* and *Ichthyophonus* harbor longer introns than most protistan eukaryotes, similar in length to those of some animals (*Figure 4B*). These similarities between ichthyosporeans and animals suggest two possible scenarios: (1) an early intronization event at the origin of Holozoa followed by reduction in some unicellular lineages (e.g., *Capsaspora* or *Corallochytrium*); or (2) independent episodes of intron proliferation in Metazoa, Choanoflagellata and Ichthyosporea. To test these hypotheses, we assembled a set of 342 paneukaryotic orthologs from 40 complete genomes and analyzed the conservation of their intron sites according to the maximum likelihood method developed by *Csűrös and Miklós (2006)* (*Figure 5—figure supplement 1*). This analysis supports the second hypothesis and reveals two independent periods of intense intron gain in unicellular holozoans: at LCA of Metazoa and Choanoflagellata, and in the branch leading to ichthyophonid Ichthyosporea (*Figure 5A–B*). After animals and choanoflagellates diverged, intron gains independently persisted in both lineages.

Our reconstruction shows that, since the origin of introns in the LECA, most ancestors were dominated by intron loss while a few remain in an equilibrium, static or dynamic (consistent with previous studies [*Csuros et al., 2011*; *Rogozin et al., 2012*]) (*Figure 5B*). A prolonged process of intron gain can be observed, however, in the lines of descent from the LECA (4.9–5.5 introns per kbp of coding sequence) to Ichthyophonida (6.9 introns/CDS kbp) and Metazoa LCAs (8.7 introns/CDS kbp), interrupted by phases of stasis with slight intron loss, such as in the Filozoa or Holozoa LCAs (*Figure 5A–B*).

The existence of independent intronization events in ancestral holozoans is supported by a hierarchical clustering analysis of the intron presence/absence profile across extant and ancestral genomes (*Figure 6A*; Ward clustering from Spearman correlation-based distances). First, most intron-rich animals form a cluster with *Salpingoeca* and *Monosiga* that also includes the LCAs of Metazoa and Metazoa + Choanoflagellata. Second, ichthyosporeans and *Corallochytrium*, although phylogenetically closely-related to each other, are highly divergent in their pattern of shared introns: the intron-dense *Creolimax* and *Sphaeroforma* form an independent cluster that differs from the Holozoa LCA; whereas *Corallochytrium* and *Chromosphaera* undergo independent secondary simplifications (from 5.5 introns/CDS kbp in the Teretosporea LCA, to 0.0 and 0.7, respectively). In contrast, *Ichthyophonus* (intron-rich) and *Capsaspora* have lower intron loss rates and are more similar to older eukaryotic ancestors, from Holozoa to the LECA (*Figure 6A*). In *Ichthyophonus*, retention is accompanied by a high gain rate, giving intron densities similar to some modern animals (7.1 intron/CDS kbp). In contrast, *Capsaspora* (3.5 intron/CDS kbp) appears to have undergone little ancestral reconfiguration of its gene architecture: there is an equilibrium between few losses and gains at the root of Filozoa (*Figure 5A*), and 85.5% of its introns are of holozoan or earlier origin (*Figure 6B*). Interestingly, introns with regulatory sites from *Capsaspora* (identified in [*Sebé-Pedrós et al., 2016b*]) have a similar, ancestral-biased, age distribution (Fisher's exact test, p-value=1; *Figure 6B*). This hints at a decoupling between the evolutionary dynamics of introns and regulatory sites, despite sharing physical space in the genome.

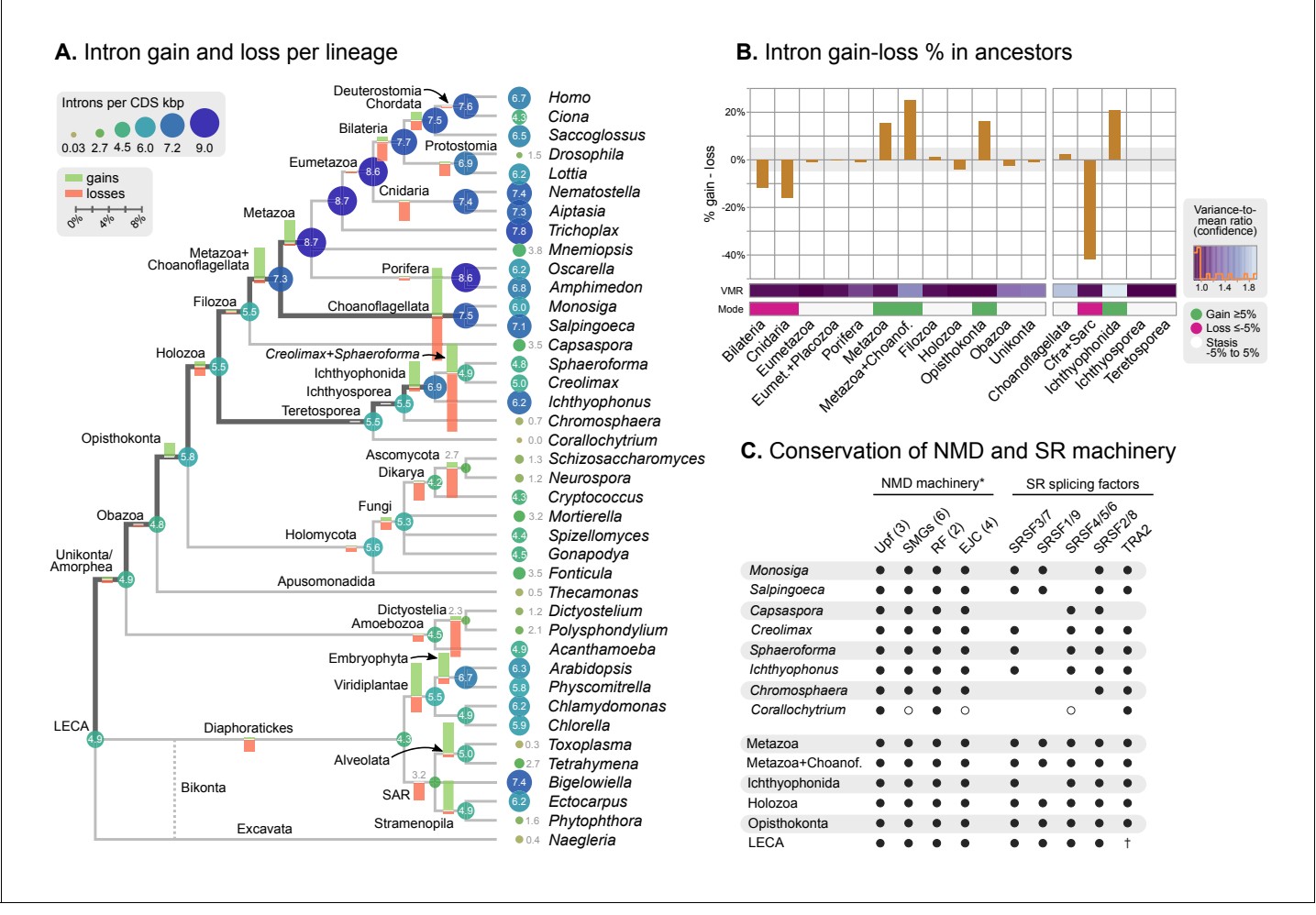

**Figure 5.** Intron evolution. (**A**) Rates of intron gain and loss per lineage, including extant genomes and ancestral reconstructed nodes. Diameter and color of circles denote the number of introns per kbp of coding sequence at each ancestral node. Bolder edges mark the lines of descent between the LECA and Metazoa/Ichthyophonida, which were characterized by continued high intron densities (see text). Red and green bars represent the inferred number of intron gains (green) and losses (red) in ancestral nodes. (**B**) Difference between intron site gains and losses in selected ancestors, including animals (left; from Metazoa to Unikonta/Amorphea) and unicellular holozoans (right). For each ancestor, we specify the variance-to-mean ratio of the inferred number of introns from 100 bootstrap replicates (higher values, denoted by lighter purple, indicate less reliable inferences; see Methods). The color code denotes modes of intron evolution: dominance of gains (green), losses (pink) and stasis (light gray). (**C**) Conservation of the NMD machinery and SR splicing factors in unicellular holozoans (up) and selected ancestors (down). Black dots indicate the presence of an ortholog, and empty dots partial conservation. For the NMD machinery, each column summarizes the presence of multiple gene families (number between brackets). † denotes the ancestral eukaryotic origin of TRA2 according to (**Plass et al., 2008**). Complete survey at the species and gene levels available as Figure 4—figure supplements 2 and 3. *Figure 5—source data 1*, *2* and *3*.

The following source data and figure supplements are available for figure 5:

**Source data 1.** Rates of gain and loss of intron sites for extant and ancestral eukaryotes, calculated for a rates-across-sites Markov model for intron evolution with branch-specific gain and loss rates (*Csurös, 2008*).

**Source data 2.** Reconstruction of intron site evolutionary histories, using a rates-across-sites Markov model for intron evolution, with branch-specific gain and loss rates (*Csurös, 2008*).

**Source data 3.** Reconstruction of the evolution of the NMD machinery (*He and Jacobson, 2015*) and key SR splicing factors (*Plass et al., 2008*).

**Figure supplement 1.** Classification of intron sites by conservation in protein alignments, as used in (*Csűrös and Miklós, 2006*; *Csurös, 2008*).

**Figure supplement 2.** Phylogenetic distribution of the NMD machinery, SR splicing factors and RNA-binding domains in eukaryotes.

*Figure 5 continued on next page*

*Figure 5 continued*

**Figure supplement 3.** Phylogenetic analysis of (**A**) eIF4A3, (**B**) Smg5/6/7, and (**C**) eRF3, using Maximum likelihood in IQ-TREE (supports are SH-like approximate likelihood ratio test/UFBS, respectively); including Bayesian inference supports for the ortologous groups of interest (BPP statistical supports, in red).

## Consequences of intron gains in early holozoan evolution

The evolutionary implications of intron gain episodes in Holozoa remain an open question. High intron densities have been linked to inefficient purifying selection: according to the mutational-hazard hypothesis, the lower effective population sizes of animals preclude the loss of slightly deleterious intronic sequence – which can constitute an impediment to genome replication or precise transcription (*Csuros et al., 2011*; *Lynch and Conery, 2003*; *Lynch, 2002*, *Lynch, 2006*). Whether this population-genetic effect is also connected with the intron gains in *Creolimax*, *Sphaeroforma* and *Ichthyophonus*, however, is unclear: their specific effective population sizes are not known, but estimates from their close relative *Sphaeroforma tapetis* are in line with typical unicellular eukaryotes (in the $10^6$ to $10^7$ range [*Marshall and Berbee, 2010*]) and thus higher than most animals (*Lynch, 2006*).

Alternatively, holozoans' intron gains could be linked to adaptive roles related to alternative splicing (AS): intron-dense genomes exhibit AS-rich transcriptomes (*Irimia and Roy, 2014*), which can

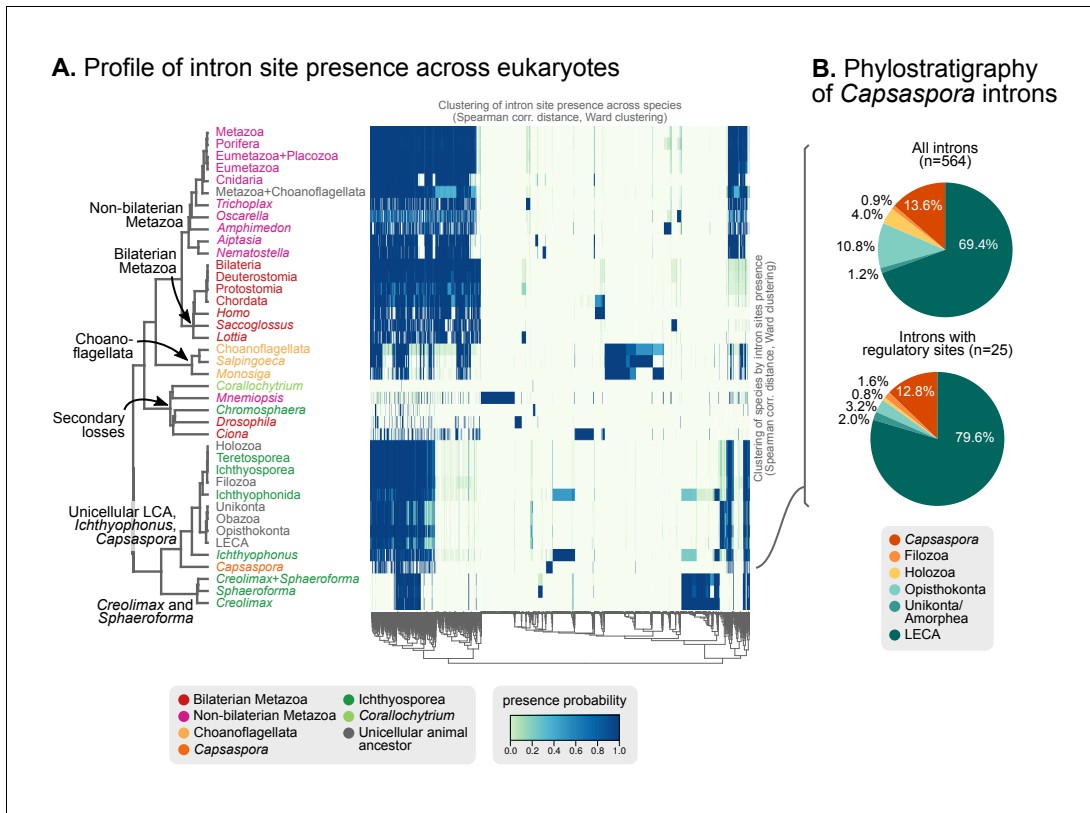

**Figure 6.** Profile of intron site presence across eukaryotes. (**A**) Heatmap representing presence/absence of 4312 intron sites (columns) from extant and ancestral holozoan genomes, plus the line of ascent to the LECA (rows). Intron sites and genomes have been grouped according to their respective patterns of co-occurrence (dendrogram based on Spearman correlation distances and Ward clustering algorithm; see Methods). The dendrogram of genome clusterings is shown to the left. *Figure 5—source data 2*. (**B**) Phylostratigraphic analysis of the origin of *Capsaspora* introns, considering all sites (left) and those with putative regulatory sites (right; after [*Sebé-Pedrós et al., 2016b*]).

increase proteomic diversity (*Barbosa-Morais et al., 2012*; *Nilsen and Graveley, 2010*; *Bush et al., 2017*) or fine-tune gene expression regulation (*Lareau et al., 2007*; *He and Jacobson, 2015*). Transcriptomes of complex animals frequently feature exon skipping events that conduce to multiple protein isoforms per gene (*McGuire et al., 2008*; *Irimia and Roy, 2014*; *Bush et al., 2017*). In contrast, the AS profiles of *Creolimax* and *Capsaspora* are dominated by intron retention (affecting 24.9% and ~33% of their genes, respectively), which can disrupt the transcripts' open reading frames (*Sebé-Pedrós et al., 2013*; *de Mendoza et al., 2015*). Intron retention is present in virtually all intron-bearing eukaryotes, pointing at an early origin in evolution (*Irimia and Roy, 2014*). Consequently, AS events in the intron-rich *Creolimax* were proposed to be involved in down-regulation of gene expression (*de Mendoza et al., 2015*) by a mechanism akin to the nonsense-mediated decay (NMD) pathway that operates in other eukaryotes (*Lareau et al., 2007*; *He and Jacobson, 2015*; *Braunschweig et al., 2014*; *Kerényi et al., 2008*).

In order to explore the relationship between intron evolution and AS-based transcriptome regulation, we surveyed the conservation in unicellular holozoans of the NMD protein complex and key splicing factors involved in AS (*Figure 5C*, *Figure 5—figure supplement 2* and *3*). The core NMD toolkit (consisting of the Upf1-3, Smg1, Smg5/6/7 and Smg8/9 genes; the release factors 1 and 3; and the exon-junction complex [EJC] [*He and Jacobson, 2015*]) has a pan-eukaryotic distribution (*Figure 5C-Figure 5—figure supplement 2A*), as previously reported for the wider spliceosomal molecular machinery (*Collins and Penny, 2005*). The NMD toolkit was also fully conserved in the LCAs of Ichthyophonida, Metazoa and Metazoa + Choanoflagellata – which underwent the above-reported intron gain episodes (*Figure 5A*). Similarly, the SR splicing factors (serine/arginine-rich proteins, termed SRSF1-9 and TRA2A/B in humans), which are involved in splice site recognition in metazoan AS (*Plass et al., 2008*; *Sanford et al., 2005*), also appeared early in eukaryotic evolution and were conserved in LCAs ranging from Opisthokonta to Metazoa (*Figure 5C*, *Figure 5—figure supplement 2B*). Interestingly, *Corallochytrium* secondarily lost part of its NMD machinery and SR splicing factors (e.g., it lacks three out of four EJC components, and only possesses one canonical SR gene) concomitantly with its acute intron losses – a process that mirrors the depletion of splicing factors in the intron-depleted ascomycete *Saccharomyces cerevisiae* (*Plass et al., 2008*). Thus, we found that the intron gain episodes of the LCAs of ichthyophonids and animals occurred in ancestral holozoans that were potentially able to perform NMD of aberrant transcripts.

## Timing of gene family diversification in holozoa

The *Monosiga* genome paper by *King et al. (2008)* revealed that much of the innovation in gene content seen in the transition to multicellularity is rooted in pervasive 'tinkering' with preexisting gene families, notably by rearrangements of protein domains. This mechanism, combined with gene duplication, allows for a functional diversification of gene families by tuning the interactions with other components of the cell—its substrate specificities, sub-cellular localization or partnerships with other proteins within larger complexes. Albeit protein domain rearrangements are not uncommon in eukaryotes (*Basu et al., 2008*, *Basu et al., 2009*; *Leonard and Richards, 2012*), this process is specifically credited with the diversification of many gene families involved in complex signaling and/or multicellular integrated lifestyle in Metazoa (*Suga et al., 2012*; *Simakov and Kawashima, 2017*; *Sebé-Pedrós et al., 2010*; *Tordai et al., 2005*; *Ekman et al., 2007*; *Hynes, 2012*; *Deshmukh et al., 2010*; *Grau-Bové et al., 2015*).

Here, we present a comprehensive study of gene diversification in Holozoa, using our taxon-rich genomic dataset to reconstruct its effect in the animal ancestry. We thus performed a comparative analysis of protein domain architectures across eukaryotes, using the rates of domain rearrangement (or shuffling) as a proxy for gene family diversification. We compared the phylogenetic distribution of protein domain co-occurrences across species and gene families (using a dataset comprising 26,377 gene families or clusters of orthologs derived from 40 eukaryotic species (see Methods). We inferred rates of domain rearrangement at ancestral nodes of the eukaryotic tree using a probabilistic birth-and-death model (*Csűrös and Miklós, 2006*) to reconstruct the content of specific protein domain architectures in ancestral genomes (available as *Figure 7*). In our approach, pairs of domains can create novel combinations ('gain') that diversify existing gene families, or dissociate domains ('loss'), which results in decreased diversity of multi-domain proteins.

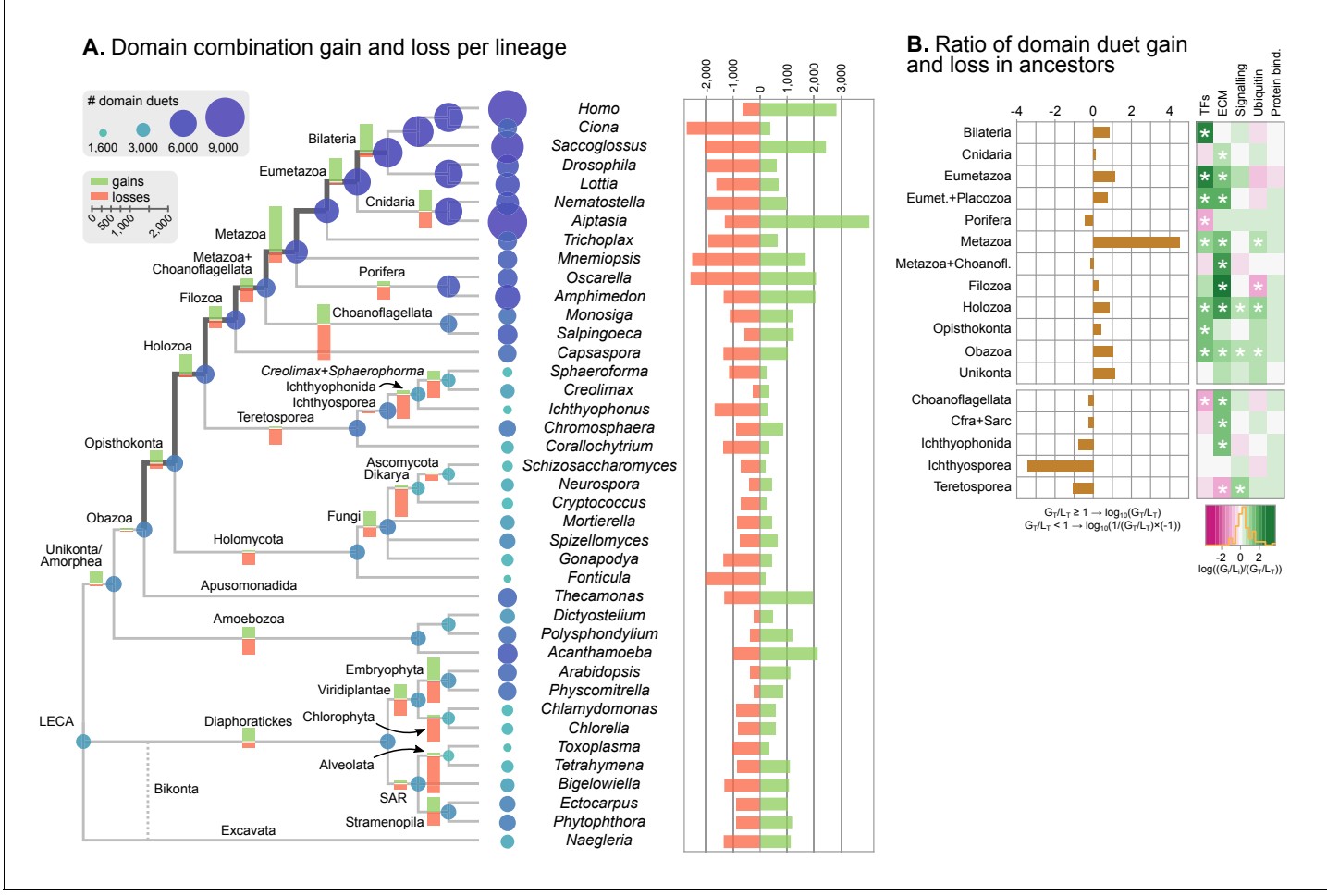

**Figure 7.** Evolution of protein domain architectures. (A) Protein domain combination gain and loss per lineage, including extant genomes and ancestral reconstructed nodes. Diameter and color of circles denote the number of different domain combinations (in different gene families) in that node of the tree. Bolder edges mark the line of descent between the LCAs of Opisthokonta and Bilateria, which was generally dominated by gains of protein domain combinations (see text). Red and green bars represent the inferred number of gains and losses, respectively. (B) Gain/loss ratio of protein domain diversity in selected ancestors, including animals (upper chart; from Metazoa to Unikonta/Amorphea) and unicellular holozoans (lower). Heatmap to the right represents the log-ratio value of the diversification rate for selected sub-sets of functionally-related protein domains relevant to multicellularity: green indicates higher-than-average diversification; pink less; white asterisks indicate two-fold or more increases or decreases (all comparisons relative to the whole set of protein domains). Source Data *Figure 7—source data 1, 2, 3* and *4*.

The following source data and figure supplement are available for figure 7:

**Source data 1.** Rates of gain and loss of protein domain pairs within a given orthogroup for extant and ancestral eukaryotes, calculated for a phylogenetic birth-and-death probabilistic model that accounts for gains, losses and duplications (*Csurös, 2010*).

**Source data 2.** Reconstruction of the evolutionary histories of protein domain pairs gains within orthogroups, using a phylogenetic birth-and-death probabilistic model that accounts for gains, losses and duplications (*Csurös, 2010*).

**Source data 3.** Reconstruction of the evolutionary histories of individual protein domains, using Dollo parsimony and accounting for gains and losses (*Csurös, 2010*).

**Source data 4.** Rates of gain and loss of orthogroups for extant and ancestral eukaryotes, using a phylogenetic birth-and-death probabilistic model that accounts for gains, losses and duplications.

**Figure supplement 1.** Gains and losses of individual protein domains across eukaryotes.

## Shuffling of protein domain architectures is common in the holozoan ancestors

We assessed the frequency of protein domain rearrangements by quantifying the rates of domain pair gain and loss at each node of the eukaryotic tree (number of gained or lost domain pairs relative to the total number of pairs in that node) (*Figure 7A–B*). Gains and losses are frequent but unequally distributed across organisms and over time, with a majority of nodes showing a tendency towards destruction or creation of domain combinations. Out of 73 analyzed organisms, 20 show a strong bias towards gains, 32 a bias towards losses (>5% difference in either sense), and 64 show combined rates of gain and loss of >10% (*Figure 7A*). In contrast, the ancestral reconstruction of individual protein domain evolution (based on Dollo parsimony) showed that losses dominate in most nodes, both extant and ancestral – with the exception of animals and their ancestors (*Figure 7—figure supplement 1*) (*Zmasek and Godzik, 2011*).

In this scenario of pervasive domain rearrangements, we identified a consistent pattern of creation of protein domain architectures in the lineage leading to Metazoa – specifically, the line of descent from the opisthokont to the bilaterian LCA (*Figure 7A–B*). This tendency was most acute at three points in animal prehistory: the Holozoa LCA, the Filozoa LCA (*Capsaspora*, animals and choanoflagellates) and the Metazoa LCA. Conversely, unicellular holozoans outside the animal lineage were dominated by secondary simplification (e.g., the LCAs of choanoflagellates or ichthyosporeans, as well as some individual species such as *Sphaeroforma*, *Ichthyophonus* or *Corallochytrium*) or by dynamic stasis (e.g., *Capsaspora*, *Creolimax* or *Chromosphaera*). Our analysis thus shows that the increased diversity of protein organizations in animals has its roots in successive events of domain shuffling during their unicellular holozoan prehistory, even if this period was dominated by a relative stasis in terms of the emergence of new protein domain families (*Figure 7A* and *Figure 7—figure supplement 1*).

Then, we questioned whether these expansions were more frequent in protein domains related to typical multicellular functions, such as the extracellular matrix (ECM), transcription factors (TF) or signaling pathways (*Suga et al., 2013*; *Richter and King, 2013*; *de Mendoza et al., 2013*; *Hynes, 2012*; *de Mendoza et al., 2014*). We found that gene families carrying TF- and ECM-related domains had consistently higher diversification rates not only in Metazoa but also in their unicellular ancestors (*Figure 7B*, right panel; asterisks indicate two-fold differences). We thus identify a continuous process of protein diversity gain involving multicellularity-related genes in animal ancestors ranging from the LCA of Obazoa (Opisthokonta + Apusomonadida) to the LCA of Metazoa.

## A unique mode of transcription factor diversification in premetazoan ancestors

Next, we analyzed the dynamics of the bursts of innovation in protein domain architectures in the unicellular ancestry of Metazoa, particularly regarding TFs and ECM-related genes. Specifically, we examined the degree of protein domain promiscuity across gene families (i.e., whether a specific domain combination is re-used in multiple gene families) in different ancestors, to measure changes in the specificity of protein domain architecture diversity.

We measured domain promiscuity by modeling each proteome as a network graph, where vertices represented protein domains that were linked by edges if they co-occurred in a given gene family (with ≥90% probability for the ancestral reconstructions; Methods and *Figure 8*). In this context, highly promiscuous domains would join multiple gene families within the network, whereas gene family-specific domains would form independent clusters. This effect can be investigated by computing the network modularity: a parameter describing the degree of isolation of 'modules' (here, groups of co-occurring domains) within a network given their connections to other 'modules' (*Figure 8C*).

We identified a general tendency for multi-domain protein families to diversify by acquisition of highly promiscuous domains also present in other families. This result was based on two observations. First, network modularities were high in most analyzed genomes (within the 0.7–1 range; consistent with previous observations (*Itoh et al., 2007*; *Xie et al., 2011*)) but they were generally lower in animals than in their unicellular relatives and ancestors (*Figure 8A*). Second, there was a strong negative relationship between modularity and the number of protein domains per gene family (Spearman's rank correlation coefficient, $\rho_s=-0.96$, p<0.001, *Figure 8B*). Therefore, at the genome

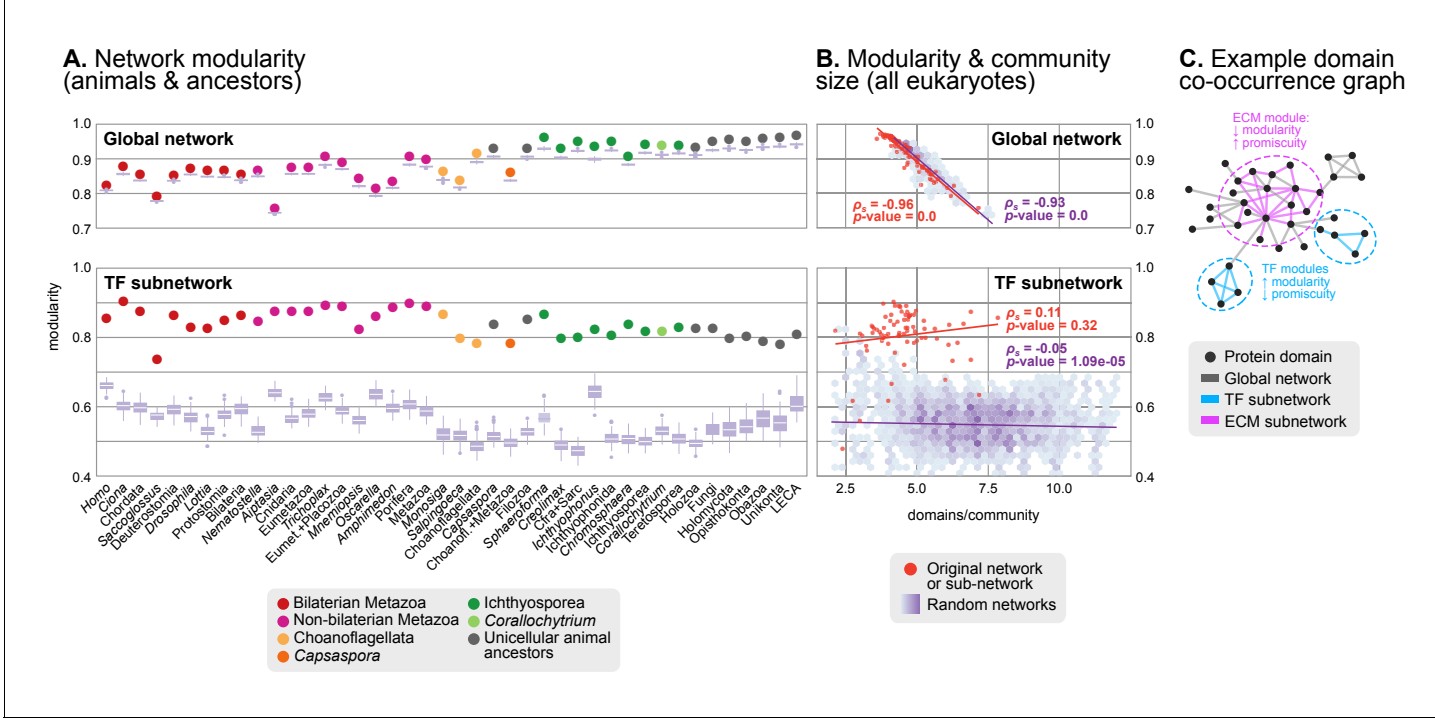

**Figure 8.** Protein domain architecture networks. (**A and B**) Modularity and community size of the global network of domain pairs (upper panels) and the TF subnetwork (lower panels), with ≥90% probability. The modularity parameter measures the fraction of the intra-community edges in the network, minus the expected value in a random network (takes values from 0 to 1; see Materials and methods and [**Newman and Girvan, 2004**]). Panels at the left show the observed modularity of the protein domain (sub)networks of various genomes (Holozoa and selected ancestors; dots are taxa-colored). Purple box plots represent the distribution of simulated modularities from 100 rewirings of the original organism-specific networks, while keeping a constant vertex degree distribution. Panels to the right show the relationship between modularities and the number of domains/community, both for actual genomes (orange) and simulated rewired networks (purple density plot, see Methods). Monotonic dependence between modularity and domains/community was tested for each set of data (global, TF and their respective simulations) using Spearman's rank correlation coefficient ($\rho_s$), and linear regression fits are included for clarity. Note that simulated TF subnetworks are less modular and have more domains/community than the original ones, signaling their higher-than-expected modularities. Note that the scales of the vertical axes change between upper and lower panels. (**C**) Example of protein domain co-occurrence network. Vertices represent domains, linked by edges if they co-occur within the same gene family. Two subnetworks are highlighted in yellow (domain pairs occurring in TF genes) or green (same for signaling genes). *Figure 7—source data 1* and *2*, *Figure 1—source data 2*.

The following figure supplement is available for figure 8:

**Figure supplement 1.** Modularity of protein domain co-occurrence networks of multicellularity-related gene sets across eukaryotes.

level, gene family diversification tends to reduce modularity due to the use of highly promiscuous protein domains, as it has been frequently reported in animals (*Simakov and Kawashima, 2017*; *Basu et al., 2008*). This same effect was observed when we analyzed subsets of the proteome networks sharing a common function: the diversification of gene families with domains related to the ECM, signaling, ubiquitination or protein–protein interactions occurs by acquisition of promiscuous domains that reduce their modularity (with $\rho_s$ in the range −0.32 to −0.84 and p<0.001; *Figure 8— figure supplement 1A–D*), and this reduction is frequently stronger in animals than in their unicellular relatives and ancestors (*Figure 8—figure supplement 1E–H*). The high promiscuity of domains mediating protein-protein interactions has already been reported in previous analyses (*Basu et al., 2008*; *Zmasek and Godzik, 2012*), thus confirming the validity of our approach.

However, the analysis of the transcription factor domain sub-networks exhibited an opposite signal: animal TF genes have more exclusive domains than their unicellular ancestors or relatives (reflected by higher modularities; *Figure 8A*, lower panel). Also, there was no negative relationship between the number of domains per community and the network modularity ($\rho_s$=0.12,

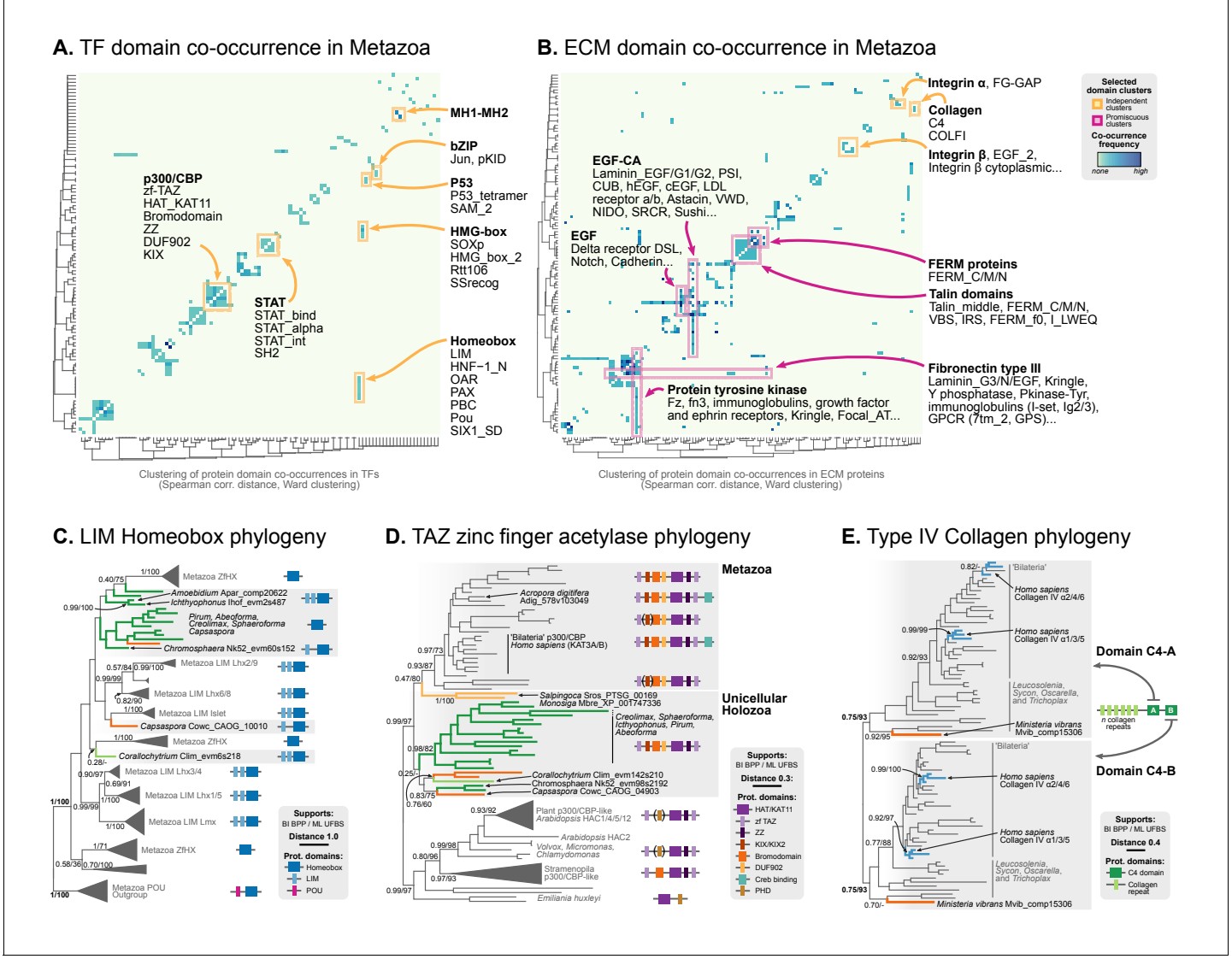

**Figure 9.** Phylogenetic analysis of the premetazoan gene families LIM Homeobox, CBP/p300 and type IV collagen. (**A and B**) Protein domain co-occurrence matrices of transcription factor (TF) (**A**) or extracellular matrix (ECM)-related gene families (**B**), inferred at the LCA of Metazoa (≥90% probability). Horizontal and vertical axes of the heatmap represent individual protein domains and their mutual co-occurrence frequency, and have been clustered according to the number of shared domains (dendrogram based on Spearman correlation distances and Ward clustering algorithm). Note that, for TFs, most co-occurrence clusters are located along the diagonal, indicating isolated domain communities; whereas ECM genes tend to contain promiscuous domains shared in multiple domain co-occurrence communities. Representative examples of independent and promiscuous domain clusters have been highlighted in both heat maps (orange and pink, respectively). (**C**) Phylogenetic tree of LIM Homeobox TFs, with mapped protein domains architectures. (**D**) Phylogenetic tree of CBP/p300 TFs based on HAT/KAT11 domain, with mapped consensus protein domain architectures. (**E**) Phylogeny of type IV collagen genes based on the C4 domain. All extant homologs, from *Ministeria* to animals, have a C4-C4 dual arrangement of filozoan origin (reflected in the phylogeny by two parallel clades representing the first and second domains within each gene). *Ministeria* (orange) and human (blue) homologs are highlighted. In C, D and E panels, bold branches represent unicellular holozoan genes and are color-coded by taxonomic assignment. All trees are Bayesian inferences (BI). Protein domain architectures and statistical supports (BPP/UFBS) are shown for selected nodes (see *Figure 9—figure supplement 1* for the complete BI and ML trees with statistical supports). *Figure 7—source data 1* and *2*.

The following figure supplement is available for figure 9:

**Figure supplement 1.** Phylogenetic analysis of the (**A**) LIM-Homeobox, (**B**) p300/CBP, and (**C**) Collagen Type IV, using Maximum likelihood in IQ-TREE (supports are SH-like approximate likelihood ratio test/UFBS, respectively) and Bayesian inference in Mr. Bayes (BPP statistical supports).

| Transcription factors | Metazoa | Metazoa + Choanoflagellata | Filozoa | Holozoa | Opisthokonta | Unikonta/ Amorphea |
|---|---|---|---|---|---|---|
| ARID | 0.022 RFX_DNA_binding, RBB1NT, DUF3518 | | | 0.214 Tudor-knot | 0.149 PLU-1 | |
| bZIP_1 | 0.048 pKID, Jun | | | | | |
| CSD | 0.254 zf-CCHC | | | | | |
| CUT | 0.198 Homeobox | | | | | |
| Ets | 0.104 SAM_PNT | | | | | |
| GATA | 0.537 BAH, ELM_2 | | | | | |
| HLH | 0.281 PAS_3, Hairy_orange | | 0.222 MITF_TFEB_C_3_N | 0.001 Response_reg, CRAL_TRIO, PAS, PAS_9, PAS_11 | | |
| HMG_box | 1.000 SOXp | | | | | |
| Homeobox | 0.001 OAR, SIX1_SD, Pou, PAX, PBC, CUT, HNF-1_N | | | 0.446 LIM | | |
| Homeobox_KN | 0.254 Meis_PKNOX_N | | | | | |
| HTH_psq | 0.168 DDE_1, HTH_Tnp_Tc5 | | | | | |
| IRF | 0.036 IRF-3 | | | | | |
| IRF-3 | 0.036 IRF | | | | | |
| LAG1-DNAbind | | | | | 0.020 BTD | |
| MH1 | 0.000 MH2 | | | | | |
| Myb_DNA-binding | 0.664 DnaJ, SWIRM-assoc_3 | | | | 0.345 RAC_head | 0.305 ZZ |
| NDT80_PhoG | | | 0.018 MRF_C1, Peptidase_S74 | | | |
| P53 | | 0.020 SAM_2 | | 0.044 P53_tetramer, SAM_1 | | |
| RFX_DNA_binding | 0.136 ARID | | | | | |
| Runt | | | | 0.030 Ank_4 | | |
| SRF-TF | | | | 0.044 HJURP_C | | |
| zf-BED | 0.281 Dimer_Tnp_hAT | | | | | |
| zf-C2H2 | 0.332 SET, zf-C2H2_4, zf-H2C2_5, zf-met, zf-H2C2_2 | | | | 0.105 zf-C2H2_6 | |
| zf-C2HC | 0.136 MOZ_SAS | | | | | |
| zf-C4 | 0.600 Hormone_recep | | | | | |
| zf-GRF | 0.537 Rnase_T | | | | | 0.305 DUF2439, AAA_12 |
| zf-MIZ | 0.071 PINIT | | | | 0.030 SAP | 0.026 PINIT |
| zf-TAZ | | | | 0.114 Bromodomain, DUF902, KIX | | |

**Figure 10.** Domain combinations that appear in transcription factor (TF) families in unicellular premetazoans, from the LCA of Unikonta/Amorphea to the LCA of Metazoa. First and second columns indicate the TF family and its inferred evolutionary origin, respectively (from [*de Mendoza et al., 2013*]). Subsequent columns list (i) the *p*-value of a Fisher's exact test for the relative enrichment of that TF family in that node of the tree (compared to other domains that rearrange there; p-values<0.05 in green); and (ii) the accessory domains that appear within each TF family. *Figure 7—source data 2*, Table 1.

The following source data is available for figure 10:

**Source data 1.** Probability of emergence of protein domain combinations present in the LCA of Metazoa in previous ancestral nodes (from LCA of Metazoa to LCA of Unikonta/Amorphea).

p-value=0.32), meaning that the addition of new domains to TF genes occurred in a gene family-specific manner (*Figure 8B*). This implies that the expanded TF repertoires of animal genomes (*de Mendoza et al., 2013*) preferentially diversify their protein domain architectures by acquiring new, not promiscuous, domains.

In summary, we identify a distinct dynamics of protein domain rearrangements for TF families in the LCA of Metazoa: new domains tend to be acquired in a family-specific manner (as opposed to reuse of promiscuous domains), contributing to the functional specialization of the animal TF repertoire.

## Gene family-specific protein domain diversification: TFs and collagen IV

Our ancestral reconstruction of protein domain architectures (*Figure 7*) allowed us to investigate the evolutionary origin of specific domain organizations within gene families and examine their diversification pattern in the ancestry of animals (Table 1). For example, we recovered many examples of gene family-specific domain diversification in novel animal TFs (Figure 10): Homeobox families (OAR, PBC/X, SIX, CUT, Pou, HNF or PAX families), TALE Homeobox (Homeobox_KN domain; Meis/Knox families), MH (MH1 and MH2 domains), bZIPs (Jun), C4 zinc finger (nuclear hormone receptors), Ets (Ets with modified SAM motifs) and HMG-box (SOX). Interestingly, the functions of accessory domains were often related to regulation of TF multimerisation or the DNA-binding affinities of the protein (*de Mendoza et al., 2013*; *Sebé-Pedrós et al., 2011*; *Holland et al., 2007*; *Holland, 2013*). These TF families appeared as isolated clusters when we sorted protein domains by their pattern of co-occurrence in the reconstructed Metazoa LCA (*Figure 9A*). Furthermore, we detected an unexpected premetazoan origin for some TF classes as per their domain combinations (*Figure 10*). We

validated two case-in-point examples by phylogenetic analysis, in order to illustrate the distinct pattern of TF domain diversification: the LIM Homeobox (LIM-HD) and p300/CBP transcriptional coactivators.

LIM homeobox genes have been classified as an animal-specific non-TALE family (*Srivastava et al., 2010b*). However, we identified LIM-associated homeobox genes in multiple ichthyosporeans, *Corallochytrium* and *Capsaspora*. We classified these candidate genes according to HomeoDB (*Zhong and Holland, 2011*) using (*Holland et al., 2007*) as a phylogenetic reference. Our analysis identified *bona fide* LIM-HD homologs with 1–2 LIM domains in *Corallochytrium*, *Chromosphaera*, *Ichthyophonus*, *Amoebidium* and *Capsaspora* (which had 1–2 LIM domains and a homeodomain); together with many LIM-devoid homologs in *Creolimax*, *Sphaeroforma*, *Pirum* and *Abeoforma* (*Figure 9C*). None of the unicellular holozoan LIM-HD genes could be confidently assigned to animal LIM homeodomain subfamilies (*Lhx1/5, Lhx3/4, Lmx, Islet, Lhx2/9, Lhx6/8*), probably because they emerged before LIM-HD radiation in animals. As such, they also predate the establishment of the LIM code of cell type specification, which has been shown to control neuronal differentiation via combinatorial expression of LIM-HD subfamilies, in animals from *Caenorhabditis elegans* to mammals or the sea walnut *Mnemiopsis* (*Simmons et al., 2012*; *Thor et al., 1999*; *Gadd et al., 2011*). Given that transcriptionally regulated cell type specification has already been demonstrated in *Creolimax* (*de Mendoza et al., 2015*), the presence of LIM-HD paralogs in ichthyosporeans will require further examination, as it raises the possibility of a conserved or convergent regulatory role in cell differentiation.

The p300/CBP TF is a transcriptional activator that contributes to distal enhancer demarcation by histone acetylation in bilaterian animals and *Nematostella* (*Gaiti et al., 2017a*). Most eukaryotes have a consensus architecture composed of a central HAT/KAT11 domain (acetylase) flanked by three zinc fingers of TAZ (2) and ZZ (1) types (DNA-binding motifs) (*Figure 9D*). Animal p300/CBP homologs typically include an additional 3-domain structure, N-terminal to the acetylase domain, composed of KIX-Bromodomain-DUF902. KIX recognizes and binds to CREB in animals (a cAMP-responsibe bZIP TF), and the Bromodomain is responsible for interaction with acetylated histones. We identified this protein domain architecture in both *Capsaspora* and ichthyosporeans, which also have the CREB gene (*Sebé-Pedrós et al., 2011*). Intriguingly, *Capsaspora*'s epigenome contains p300/CBP-specific histone acetylation marks, but its relatively compact genome lacks distal enhancers (*Sebé-Pedrós et al., 2016b*).

Finally, in stark contrast to TF domain-specific diversifications, clusters of co-occurring protein domains in ECM-related genes were dominated by highly promiscuous domains shared between different gene families (*Figure 9B*). This pattern explains the lower network modularity of animal ECM genes (*Figure 8—figure supplement 1*). Among the most promiscuous domains, we found epidermal growth factor-related domains (EGF-CA, EGF), type III fibronectin or protein tyrosine kinase motifs, consistent with previous observations (*Cromar et al., 2014*). These domains are part of multiple, functionally different gene families: structural laminins, immunoglobulins, the Notch/Delta signaling system, LDL receptors or GPCR signaling genes (pink highlight, *Figure 9B*).

The diversification of collagen genes, however, is a counterexample to the promiscuous domain shuffling at the ECM: like many TFs, collagens typically contain repetitive motifs with unique domains conferring functional specificity (*Hynes, 2012*). This includes, for example, structural fibrillar collagens (COLFI domains and further specialization within metazoans), type XV/XVIII (endostatin/NC10 domains), type IV collagen or type IV-like spongins (specific of invertebrate metazoans); there are also non-structural genes like collectin receptors (Lectin-C) or the C1q complement subcomponent (C1q) (*Hynes, 2012*; *Aouacheria et al., 2006*; *Heino, 2007*; *Fahey and Degnan, 2012*; *Exposito et al., 2008*). Most collagen genes appeared and expanded in Metazoa, concomitantly with the ECM structures they associate with (*Hynes, 2012*; *Fidler et al., 2017*). We found, however, a remarkable exception: a canonical type IV collagen gene in the filasterean *Ministeria vibrans*, a naked filose amoeba devoid of basement membrane or ECM (*Patterson et al., 1993*; *Cavalier-Smith and Chao, 2003*). Cross-linked type IV collagens are part of the structural core of animal basement membranes (to date, all of its components had been described as exclusive to animals) (*Hynes, 2012*; *Fidler et al., 2017*). This *Ministeria* ortholog is composed of a pair of C4 domains at the C-terminus and multiple collagen Gly-X-Y repeats. Phylogenetic analysis of C4 showed that this domain arrangement appeared from two duplicated motifs within the same protein, and its order is thoroughly conserved in animals and *Ministeria* (*Figure 9E*). Thus, a canonical type IV collagen was

already present in the common ancestor of filastereans, choanoflagellates and animals – which was unicellular and most likely lacked ECM or basement membrane-like structures. The essential role of collagen IV in the organization of extant metazoans' tissues (*Fidler et al., 2017*) would therefore require a co-option from an earlier function in a unicellular context, as it has been previously proposed for other ECM components such as the integrin adhesome (*Sebé-Pedrós et al., 2010*) or cadherins (*Abedin and King, 2008*).

## Discussion

We have investigated the evolutionary dynamics of key genomic traits in the unicellular ancestry of Metazoa, in the first comparative genomic study that simultaneously includes all unicellular holozoan lineages, and more than one species per lineage: animals, seven Teretosporea genomes (six ichthyosporeans and *Corallochytrium*), *Capsaspora*, and two choanoflagellates (*Salpingeoca* and *Monosiga*). Our enhanced taxon sampling, including four newly sequenced genomes, allows us to perform both within- and across-lineage comparisons, thus covering the different time scales at which the evolution of coding and non-coding genome features occurred.

### Dating the origin of animal-like protein domain architectures, intron architecture and genome size

We have identified continued process of gene innovation in terms of protein domain architectures in the animal ancestry, peaking at the LCA of Holozoa. This burst of diversification, enriched in TFs and ECM-related domains (*Figure 7B*), set the foundations of the animal-like gene tool-kits of unicellular holozoans that have been reported in previous studies of gene family evolution regarding signaling pathways (*Suga et al., 2012*; *Grau-Bové et al., 2015*, *2013*), cell adhesion systems (*de Mendoza et al., 2015*; *Nichols et al., 2012*; *Sebé-Pedrós et al., 2010*) and transcription factors, often involved in developmental processes (*de Mendoza et al., 2013*; *Sebé-Pedrós et al., 2011*). The expansion of protein diversity in early holozoans provided fertile ground for the frequent co-option of ancestral genes for multicellular functions in Metazoa (*Richter and King, 2013*). Overall, our probabilistic reconstruction of the genome content of unicellular animal ancestors (available as Figure 7—source data 7) provides a useful framework for targeted analysis of gene evolution and protein domain architecture evolution. As case-in-point examples of our approach, we have established the premetazoan origin of the transcription factors LIM Homeobox (present in Ichthyopsorea and *Capsapsora*) and p300/CBP-like (all unicellular Holozoa) (*Figure 9C–E*), and canonical Type IV collagens, a key element of the animal ECM (*Hynes, 2012*) (present in the filasterean amoeba *Ministeria vibrans*).

We have also investigated the time of origin of intron-rich genomes in Holozoa. We detect three independent episodes of massive intron gain: (1) at the root of Metazoa, (2) the shared LCA between Metazoa and Choanoflagellata, and (3) the root of ichthyophonid Ichthyosporea (*Creolimax*, *Sphaeroforma* and *Ichthyophonus*). Furthermore, since the early origin of introns in the earliest eukaryotes (*Irimia and Roy, 2014*), the ancestry of both animals and ichthyophonids maintained a state of high intron density. The evolutionary implications of this circumstance, however, remain an open question. First, the independent intron gain episodes of animals and unicellular holozoans are mirrored by two different modes of alternative splicing dominating in each clade: animal transcriptomes are rich in isoform-producing exon skipping (*McGuire et al., 2008*; *Irimia and Roy, 2014*), whereas most of the alternatively spliced transcripts of *Capsaspora* (*Sebé-Pedrós et al., 2013*) and *Creolimax* (*de Mendoza et al., 2015*) originate by intron retention and are thus more similar to the putative ancestral eukaryote than to Metazoa (*Irimia and Roy, 2014*). Second, we here show that the holozoan LCAs that underwent intron invasions (Ichthyophonida, Metazoa and Metazoa + Choanoflagellata) all possessed the essential NMD machinery and a rich complement of assisting splicing factors (*Figure 5C*). Thus, they were in principle able to reduce the costs imposed by slightly deleterious intron invasions, as predicted by the mutational-hazard hypothesis (*Lynch and Conery, 2003*; *Lynch, 2002*, *Lynch, 2006*). And third, the protracted state of high intron density in the ancestry of Metazoa and Ichthyophonida could have contributed to maintaining high levels of transcriptome variability that could in turn be co-opted for potentially adaptive, regulated AS events (*Irimia and Roy, 2014*; *Koonin et al., 2013*). However, we cannot at present elucidate the relative importance of adaptation and population-genetic effects in the holozoans' intron gain episodes: further

transcriptomic analyses of unicellular holozoans are required to confirm that intron retention is their ancestrally prevalent AS mode (*Sebé-Pedrós et al., 2013*; *Irimia and Roy, 2014*; *de Mendoza et al., 2015*); and the scant data on unicellular holozoans' population genetics hampers the interpretation of genome architecture evolution under the light of the mutational-hazard hypothesis (*Lynch and Conery, 2003*; *Lynch, 2002*).

We also addressed the evolution of genome size across holozoans. The emergence of larger genomes in Metazoa cannot be explained solely by intron gain and gene family expansion (*Elliott and Gregory, 2015a*). Unfortunately, other factors such as the contribution of TE invasions (*Figure 3B*) or the extension of intron sites are not possible to date at the holozoan-wide evolutionary scale due to the lack of conserved signals. A possible way out of the conundrum is to study the conserved functions in the non-coding parts of the genome. For example, the compact genome of *Capsaspora* (median intergenic regions: 373 bp) has intragenic *cis*-regulatory elements key to its temporal regulation of cell differentiation (*Sebé-Pedrós et al., 2016b*), but the putative regulatory functions in the larger intergenic regions of *Creolimax*, *Sphaeroforma* and *Salpingoeca* (median intergenic 900–1200 bp) remain uncharacterized. It is tantalizing to note that (1) *Creolimax* and *Salpingoeca* exhibit temporal differentiation of cell types (*Fairclough et al., 2013*; *de Mendoza et al., 2015*), and (2) their intergenic median sizes are in line with those of *Amphimedon* (885 bp) (*Figure 1—source data 1*), a demosponge with bilaterian-like promoters and enhancers, including distal regulation (*Gaiti et al., 2017a*, *Gaiti et al., 2017b*). However, the ancestral gene linkages conserved across Metazoa, frequently due to common *cis*-regulation (*Irimia et al., 2012*), appear to be animal innovations absent in unicellular holozoans (*Figure 3—figure supplement 1*). We thus propose that homologous regulatory regions would be rarely conserved between animals and unicellular holozoans; and only common *types* of regulatory elements could be expected, e.g. distal enhancers or developmental promoters.

## Independence of genome features in premetazoan evolution

Overall, our results show that extant holozoan genomes have been shaped by both differential retention of ancestral states and secondary innovations, for the multiple genomic traits analyzed here, namely genome size, intron density, synteny conservation, protein domain diversity and gene content (reviewed in (*Richter and King, 2013*)). We can thus conclude that the genomes of unicellular premetazoans were shaped by independent evolutionary pressures on different traits, as has been seen in Metazoa (*Simakov and Kawashima, 2017*).

Our findings can help to delimit the implicit trade-offs of choosing a unicellular model organism for functional and comparative studies with Metazoa, taking into account the loss of animal-like genomic traits relevant to different analyses. For example, phylogenetic distances between orthologous genes are shorter between some ichthyosporeans and animals than between choanoflagellates and animals (*Figure 3D*), yet choanoflagellates are more similar to the animal ancestor in terms of intron structure (*Figure 6A*) and have lower rates of protein domain diversity loss (*Figure 7B*). Interestingly, *Capsaspora* emerges as a well-suited model with a slow pace of genomic change attested for multiple traits: intron evolution, coding sequence conservation, gene order and (possibly) genome size. Its remarkable microsynteny conservation with *Corallochytrium* and *Chromosphaera* indicates the existence of ancestral holozoan gene linkages that have been disrupted, and rewired, in extant choanoflagellates, ichthyosporeans and animals (*Figure 3C*). However, *Capsaspora*'s lack of close sister groups hampers comparative studies of faster-evolving genomic features, be it the regulatory circuitry (*Sebé-Pedrós et al., 2016b*), or co-option of genetic tool-kits for its unique aggregative development (*Sebé-Pedrós et al., 2013*).

The new genomes from Ichthyosporea and *Corallochytrium* analyzed here provide novel insights into the reconstruction of premetazoan genomes. The Teretosporea clade has a deeper sampling than other unicellular holozoans and exhibit a mixture of slow- and fast-evolving genomic traits, which provides novel insights into the independence of genomic characters during premetazoan evolution. For example, *Ichthyophonus* tends to retain the ancestral intron/exon structure (*Figure 6A*) and is relatively similar to animals in terms of coding sequence conservation (*Figure 3D*), but it harbors a secondarily expanded genome with disrupted gene linkage (*Figure 3A, C*). Another example is *Corallochytrium* and *Chromosphaera*, both with massive simplifications of intron content (*Figure 5A*), but higher synteny conservation with the inferred ancestral Holozoa (*Figure 3C*). Also, the diversity of protein domain combinations of *Chromosphaera* is the highest

among ichthyosporeans (in line with values of animals and holozoan ancestors; *Figure 7A*) and phylogenetic distances to animal orthologs are comparatively low (*Figure 3D*). These studies of genome history in holozoans are key to our interpretation of functional genomics analyses. For example, *Creolimax* and *Sphaeroforma* are close species with a broadly conserved life cycle (*Glockling et al., 2013*), and they could therefore be an apt model to test hypotheses of cell type evolution in Holozoa – for example, whether new cell types emerge as lineage-specific transcriptomic specializations, as proposed by (*de Mendoza et al., 2015*). This investigation would benefit from taking into account their high microsynteny when analyzing co-regulated gene modules, while considering that *Sphaeroforma*'s multiple TE invasions could blur the conservation of non-coding regulatory elements in the intergenic regions (*Figure 3A–C*).

## Genomic innovation in the animal ancestry

Our analysis of ten unicellular holozoans has uncovered the timing of genome evolution in the ancestry of Metazoa, at both the architectural and gene content levels. In particular, we have observed that holozoan genomes evolved under temporally uncoupled dynamics for synteny reorganization, intron gains, TE propagation, coding sequence conservation and gene family diversification. Some of these traits have independent effects in extant holozoans, e.g., different episodes of intron gain or genome expansion in ichthyosporeans and animals. Yet, other traits exhibit conserved dynamics across the unicellularity/multicellularity divide: the diversification of ECM and TF gene families—including molecular tool-kits essential for multicellularity—extends back to the LCA of Holozoa; and the high intron densities in premetazoans suggest a continued state of transcriptome variability, co-optable for regulation or protein innovation, in the unicellular prehistory of Metazoa. Overall, our timeline of holozoan genome evolution offers a framework to investigate when and how premetazoan genomic elements—gene tool-kits, linkages and structure, and the non-coding sequences that harbor epigenomic regulatory elements—were functionally co-opted in multicellular animals.

## Materials and methods

### Cell cultures

*Corallochytrium limacisporum*, *Abeoforma whisleri* and *Pirum gemmata* were grown in axenic culture in marine broth medium (Difco 2216) at 18°C (*Abeoforma* and *Pirum*) or 23°C (*Corallochytrium*). *Chromosphaera* was grown in axenic culture at 18°C in YM medium (containing 3 g yeast extract, 3 g malt extract, 5 g bacto peptone, 10 g dextrose, 14.5 g Difco agar, and 25 g sodium chloride, per liter of distilled water).

### DNA and RNA extraction and sequencing

DNA-seq data was produced for *Pirum*, *Abeoforma*, *Chromosphaera* and *Corallochytrium*, by sequencing paired-end (PE) and Nextera mate-pair (MP) libraries. DNA extractions were performed from confluent axenic cultures, grown in three flasks of 25 ml for 5 days. DNA was extracted using a standard protocol by which cells were lysed in the extraction buffer composed of Tris-HCL, 50 mM EDTA, 500 mM NaCl and 10 mM ß-mercaptoethanol. DNA was purified with phenol:chloroform:iso-amyl alcohol (25:24:1) and treated with of Rnase A (Sigma Aldrich, Saint Louis, MO, USA). For each library, the read numbers, lengths and insert/fragment sizes were as follows: *Pirum*, PE 125 bp ($250 \cdot 10^6$ reads, 250 bp insert size), MP 50 bp ($108 \cdot 10^6$ reads, 6 kb fragment size); *Abeoforma*, PE 100 bp ($73 \cdot 10^6$ reads, 600 bp insert size), MP 100 bp ($41 \cdot 10^6$ reads, 6 kb fragment size); *Chromosphaera*, PE 125 bp ($143 \cdot 10^6$ reads, 250 bp insert size), MP 50 bp ($114 \cdot 10^6$ reads, 5 kb fragment size); and *Corallochytrium*, PE 100 bp ($150 \cdot 10^6$ reads, 420 bp insert size), MP 100 bp ($47 \cdot 10^6$ reads, 3 kb fragment size). All PE and MP libraries were prepared and sequenced at the CRG Genomics Unit (Barcelona), using Illumina HiSeq 2000 and the Trueseq Sequencing Kit v3 (*Abeoforma* and *Corallochytrium*) or v4 (*Pirum* and *Chromosphaera*). The only exception was *Corallochytrium* PE libraries, which were sequenced at the Earlham Institute Genomics Unit (Norwich, UK) using Illumina MiSeq and the Trueseq protocol v2. Genome sequencing data has been deposited in NCBI SRA under the BioProject accession PRJNA360047.

RNA-seq data was produced for *Chromosphaera* and *Abeoforma*. RNA extractions were performed from confluent axenic cultures grown in three 25 ml flasks for 5 days. RNA was extracted

using Trizol reagent (Life Technologies, Carlsbad, CA, USA) with a further step of Dnase I (Roche) to avoid contamination by genomic DNA, then purified using RNeasy columns (Qiagen). We sequenced PE libraries of 125 bp with an insert size of 250 bp, yielding $168 \cdot 10^6$ reads for *Chromosphaera* and $178 \cdot 10^6$ for *Abeooforma*; which were constructed using the Trueseq Sequencing Kit v4 (Illumina, San Diego, CA). The libraries were sequenced in one lane of an Illumina HiSeq 2000 at the CRG genomics unit (Barcelona). All transcriptome sequencing data has been deposited in NCBI SRA using the BioProject accession PRJNA360056.

## Genome assembly

Genomic PE and MP libraries were quality-checked using FastQC v0.11.2 (*Andrews, 2014*) and trimmed accordingly with Trimmomatic v0.33 (*Bolger et al., 2014*) to remove remnant adapter sequences (*ad hoc*) and the low-quality 5' read ends (sliding window = 4 and requiring a minimum Phred quality = 30). A minimum length equal to the original read length was required. During the quality-trimming process, libraries of unpaired forward reads were kept as single-end reads (SE). After trimming, the read survival rate for each DNA library was as follows: *Pirum,* PE 30.2%, MP 91.2%; *Abeoforma*, PE 75.5%, MP 31.0%; *Chromosphaera*, PE 81.1%, MP 89.9%; and *Corallochytrium*, PE 94.7%, MP 73.1%.

Genome assemblies were performed using Spades v3.6.2 (*Nurk et al., 2013*) with the BayesHammer error correction algorithm (*Nikolenko et al., 2013*). For each organism, PE data were analyzed using Kmergenie (*Chikhi and Medvedev, 2014*) to determine the optimal k-mer length for the assembly process, which was used in the Spades assembly in combination with smaller and larger values, including the maximum possible odd length below the maximum read length after trimming. The optimized assemble parameters for each genome were as follows: *Pirum*, max. read length = 125, k = 55,123; *Abeoforma*, max. read length = 100, k = 47,91; *Chromosphaera*, max. read length = 125, k = 91,121; *Corallochytrium*, max. read length = 100, k = 41,63,91. In the cases of *Corallochytrium* and *Chromosphaera* genomes, Spades was run in *careful* mode, taking into account PE, SE and MP data in the same run. In the cases of the highly repetitive *Abeoforma* and *Pirum* genomes, an initial Spades assembly of PE and SE libraries was combined with MP libraries using the Platanus v1.2.1 scaffolding module (*Kajitani et al., 2014*). Each assembly was later processed using the GapCloser module from SOAPdenovo assembler with PE data, in order to extend the scaffolded contigs by shortening N stretches (*Luo et al., 2012*). Genome assembly statistics (genome size, N50, L75) were calculated using Quast v2.3 (*Gurevich et al., 2013*), and completeness was assessed using the BUSCO v1.1 (*Simão et al., 2015*) database of universal eukaryotic genes, based on the predicted transcripts.

## Genome annotation

Genome feature annotations were produced for *Corallochytrium*, *Chromosphaera*, *Abeoforma*, *Pirum* and *Ichthyophonus.* We used evidence-based gene finders (relying on transcript/peptide mapping: Augustus v3.1 (*Keller et al., 2011*) and PASA v2.0.2 [*Haas et al., 2003*, *2008*]), plus complementary *ab initio* predictors (based on hidden Markov models for gene structure: GeneMark-ES v4.21 (*Lomsadze et al., 2005*) and SNAP [*Korf, 2004*]). These results were combined to produce a consolidated gene annotation using Evidence Modeler v1.1.1 (*Haas et al., 2008*).

SNAP and GeneMark-ES annotations were iterated for three times on the final genome assemblies, using the output of each step as a training set for the next one (the first SNAP prediction was done using the standard minimal HMM; GeneMark-ES was omitted for *Abeoforma* and *Pirum* due to its highly fragmented gene bodies, which impaired intron delimitation).

Transcriptome assemblies were produced to support PASA and Augustus gene predictions. RNA-seq PE libraries were assembled using genome-guided Trinity v2.0.6 and STAR v2.5 (for *Corallochytrium*, *Chromosphaera* and *Ichthyophonus*) or de novo Trinity (*Pirum* and *Abeoforma*, assemblies from (*Torruella et al., 2015*; *Grabherr et al., 2011*; *Dobin and Gingeras, 2015*). In the case of the *Corallochytrium*, *Chromosphaera* and *Ichthyophonus* genome-guided assemblies, quality control was performed as indicated above for the genomic libraries, using the RNA-seq data generated for this study (*Chromosphaera*) or in (*Torruella et al., 2015*) (*Ichthyophonus* accession: PRJNA264423; *Corallochytrium* accession: PRJNA262632). A minimum k-mer coverage = 2 was used in all Trinity assemblies. Transcriptome assemblies were annotated with Transdecoder using Pfam release 29

protein domain database, in order to obtain mRNA and translated peptides. Next, PASA annotations were obtained from assembled transcripts, mapped to the genome using GMAP and BLAT v35 (*Kent, 2002*; *Wu et al., 2016*). Only high quality mapping was accepted, with a minimum of 95% identity and 75% transcript coverage. We then trained Augustus independently, using protein and mRNA predictions (mapped to the genome with Scipio 1.4 (*Keller et al., 2008*), BLAT and GMAP), followed by an optimization round of the species-specific parameters. After the training, an Augustus prediction was performed using the optimized parameters.

Finally, all annotations were consolidated using Evidence Modeler. In this step, gene models from PASA and Augustus were given higher relative weights than *ab initio*-predicted models (10 and 5 times more reliability, respectively).

## Phylogenomic analysis

We used an improved version of the dataset published by Torruella *et al.* (*Torruella et al., 2015*), adding nine single-copy protein domains to the previous version (which included 78 alignments) according to the methodology developed in (*Torruella et al., 2012*). Since *Abeoforma* and *Pirum* genome assemblies were fragmented and contained partial gene models, we used transcriptome assemblies from (*Torruella et al., 2015*) instead. We compiled a 57-taxa dataset of Unikonta/Amorphea species (hereby termed BVD57 taxa matrix; including Holozoa, Holomycota, Breviatea, Apusomonadida and Amoebozoa; 24,021 amino acid positions). This dataset represents a ~ 10% increase in the number of aligned positions, compared to the original S70 dataset from (*Torruella et al., 2015*).

We used the BVD57 dataset to build ML phylogenetic trees using IQ-TREE v1.5.1 (*Nguyen et al., 2015*), under the LG model with a 7-categories free-rate distribution, and a frequency mixture model with 60 frequency component profiles based on CAT (LG + R7+C60) (*Quang et al., 2008*). LG + R7 was selected as the best-fitting model according to the IQ-TREE *TESTNEW* algorithm as per the Bayesian information criterion (BIC), and the C60 CAT approximation was added because of its higher rate of true topology inference (*Quang et al., 2008*). Statistical supports were drawn from 1000 ultrafast bootstrap values with a 0.99 minimum correlation as convergence criterion (*Minh et al., 2013*) and 1000 replicates of the SH-like approximate likelihood ratio test (*Guindon et al., 2010*), for all models stated above. Furthermore, 500 non-parametric bootstrap replicates were computed for the LG + R7+PMSF CAT approximation (as this was the only CAT approximation for which non-parametric bootstraps could be calculated in a feasible computation time).

We then used the same alignment to build a Bayesian inference tree with Phylobayes MPI v1.5 (*Lartillot et al., 2013*), using the LG exchange rate matrix with a 7-categories gamma distribution and the non-parametric CAT model (*Lartillot and Philippe, 2004*) (LG+$\Gamma$7 + CAT). A $\Gamma$7 distribution was considered to be the closest approximation to the free-rates R7 distribution of the IQ-TREE ML analysis (as free-rates distributions are not implemented in Phylobayes). We removed constant sites to reduce computation time. We ran two independent chains for 1231 generations until convergence was achieved (maximum discrepancy <0.1) with a burn-in value of 32% (381 trees). The adequate burn-in value was selected by sequentially increasing the number of burn-in trees, until we achieved (1) a minimum value of the maximum discrepancy statistic, and (2) the highest possible effective size for the log-likelihood parameter. The *bpcomp* analysis of the sampled trees yielded a maximum discrepancy = 0.095 and a mean discrepancy = 0.001. The *tracecomp* parameter analysis gave an effective size for the log-likelihood parameter = 37; and the minimum effective size = 11 (for the alpha statistic).

## Generation of a species tree and ortholog datasets for comparative analyses

Our comparative genomics analyses are based on a dataset of 42 complete eukaryotic genomes, with a focus on unicellular and multicellular Holozoa, and using relevant outgroups from the Holomycota, Apusomonadida, Amoebozoa, Viridiplantae, Stramenopila, Alveolata, Rhizaria and Excavata groups. The complete list of species, abbreviations and data sources is available as *Figure 2—source data 1*.

Since ancestral state reconstruction requires the assumption of an explicit species tree, we classified the 42 genomes in our dataset according to a consensus of phylogenomic studies (*Torruella et al., 2015*; *Derelle et al., 2015*; *He et al., 2014*) and our own results. We remained agnostic about the internal topology of SAR (*Burki et al., 2016*), Fungi (*Torruella et al., 2015*), the contentious hypotheses for the root of eukaryotes (namely, 'Opimoda-Diphoda' or 'Excavata-first') (*Derelle et al., 2015*; *He et al., 2014*) and the earliest-branching animal group (Porifera or Ctenophora) (*Whelan et al., 2015*; *Simion et al., 2017*). All these cases were recorded in our species tree as polytomic branchings.

We inferred two different ortholog datasets using the predicted proteins from the afore-mentioned genomes, using Orthofinder v0.4.0 with a MCL inflation = 2.1 (*Emms and Kelly, 2015*). The first database included 40 eukaryotic species (excluding the low-quality gene models of *Pirum* and *Abeoforma*), whose genes were classified in 162,559 clusters of orthologs, 26,377 of which contained >1 gene (henceforth, 'orthocluster'). The second database included all available unicellular holozoan genomes (*i.e.*, six ichthyosporeans, two choanoflagellates, *Corallochytrium* and *Capsaspora*) and yielded 58,516 orthoclusters, 11,925 of which contained >1 gene.

## Gene family evolution analyses

### Retrieval of homologous sequences

Retrieval of homologous protein sequences was performed by querying orthologs or protein domain HMM profiles (depending on the gene family; see below) against a database of protein sequences from 69 selected eukaryotic genomes and transcriptomes (*Figure 2—source data 1*). Since *Abeoforma* and *Pirum* genome assemblies were fragmented and contained broken gene models, we used transcriptome assemblies from (*Torruella et al., 2015*) instead.

The following gene families were defined by its catalytic/representative protein domain: type IV collagen (PF01413), TAZ zinc finger TFs with HAT/KAT11 domains (PF08214), Upf1 (PF09416), Upf2 (PF04050), Upf3 (PF03467), Smg1 (PF15785), Smg8/9 (PF10220), eRF1 (combination of PF03463 +PF03464+PF03465), Y14 (PF09282), Magoh (PF02792) and MLN51/CASC3 (PF09405). Homologs were thus retrieved by querying Pfam protein domains (29th Pfam release (*Punta et al., 2012*)), using HMMER v3.1b2 (*HMMER, 2015*) searches with *hmmersearch*, using the profile-specific gathering threshold cut-off.

In the case of LIM homeodomain genes, we queried the genomes/transcriptomes of all available unicellular holozoans (see taxon sampling above) using the homeobox HMM (PF00046), and restricted the subsequent phylogenetic analysis (see below) to sequences that clustered with known LIM-HD genes from the HomeoDB database in *blastp* searches (*Zhong and Holland, 2011*).

In the case of the eRF3, eIF4A3, Smg5/6/7 and SRSF1-9 gene families, we queried the genomes/transcriptomes mentioned above using *blastp* searches of the human orthologs of these gene families (Uniprot accession numbers: eRF3 is P15170; eIF4A3 is P38919; Smg5/6/7 are Q9UPR3/Q86US8/Q92540; SRSF1-9/TRA2 (*Plass et al., 2008*) are Q07955, Q01130, P84103, Q08170, Q13243, Q13247, Q16629, Q9BRL6, Q13242 and P62995). For eRF3 and eIF4A3 searches, we also included a selection of orthologs from the nearest outgroup gene families: EF1-alpha and HBS1L genes for eRF3 (human accessions: Q05639/P68104 and Q9Y450); and eIF4A1/2 for eIF4A3e (human accessions: P60842/Q14240).

### Protein alignments and phylogenetic analyses (LIM homeobox, type IV collagen and CBP/p300, Smg5/6/7, eIF4A3, eRF3, SRSF1-9/TRA2 splicing factors)

Protein alignments were built with MAFFT v7.245 (*Katoh and Standley, 2013*), using the G-INS-i algorithm optimized for global homology for single-domain alignments (LIM homeobox, type IV collagen, CBP/p300 and SRSF1-9/TRA2) or the E-INS-i for multiple local homology for whole-protein alignments (Smg5/6/7, eIF4A3 and eRF3). All alignments were run for up to $10^6$ cycles of iterative refinement. Then, the resulting alignments were manually examined, curated and trimmed (a process that included the removal of non-homologous amino acid positions and, eventually, non-essential sequences containing too few aligned positions that could disrupt the subsequent phylogenetic analysis). If necessary, the alignment and trimming process was repeated to incorporate the changes from manual curation.

Phylogenetic analyses were performed in the final, trimmed alignments using two independent approaches: maximum likelihood using IQ-TREE v1.5.1 (*Nguyen et al., 2015*) and Bayesian inference using MrBayes v3.2.6 (except in the case of SRSF1-9/TRA2, in which Bayesian inference it was omitted due to the large number of retrieved sequences) (*Ronquist and Huelsenbeck, 2003*). The optimal evolutionary models for each alignment were selected using ProtTest v3.4's BIC criterion (*Darriba et al., 2011*), yielding LG+Γ4 + i as the best model for the Collagen IV, HAT/KAT11, LIM Homeobox, eRF3 and eIF4A3 phylogenies; LG+Γ4 for SRSF1-9/TRA2; and LG+Γ4 + F + i for Smg5/6/7.

For IQ-TREE (*Nguyen et al., 2015*) analyses, the best-scoring ML tree was searched for up to 100 iterations, starting from 100 initial parsimonious trees; statistical supports for the bipartitions were drawn from 1000 ultra-fast bootstrap (*Minh et al., 2013*) replicates with a 0.99 minimum correlation as convergence criterion, and 1000 replicates of the SH-like approximate likelihood ratio test. For MrBayes analyses, we ran two independent runs of four chains each (three cold, one heated) for a variable number of generations until run convergence was achieved (at different values depending on the gene family), sampling every 100 steps and running a diagnostic convergence analysis every 1000 steps. Convergence was deemed to occur when, using a 25% relative burn-in value, the average standard deviation of split frequencies was <0.01. Final number of generations for each gene family: $7.2 \cdot 10^7$ generations for Collagen IV; $1.2 \cdot 10^7$ for LIM Homeobox; and $9.9 \cdot 10^6$ for HAT/KAT11; $6.4 \cdot 10^7$ for Smg5/6/7; $1.6 \cdot 10^7$ for eRF3; $7 \cdot 10^6$ for eIF4A3.

## Other ortholog searches (Upf1, Upf2, Upf3, Smg1, Smg8/9, eRF1, Y14, Magoh and MLN51)

The following gene families, part of the NMD machinery, are unambiguously defined by the presence of their defining protein domains (see above): Upf1, Upf2, Upf3, Smg1, Smg8/9, eRF1, Y14, Magoh and MLN51. Thus, presence of the protein domain in a given species was used to establish the presence of the corresponding ortholog.

## Analysis of repetitive elements

Repetitive regions were annotated in Holozoa genomes using RepeatMasker open-4.0.5 (*Smit et al., 2015*) and annotations from the 20150807 release of GIRI RepBase database (*Bao et al., 2015*). We used the Eukaryota-specific database, with either the slow high-sensitivity search mode (unicellular holozoans) or the default search mode (metazoans); and stored the genome coordinates of TEs, low complexity repeats, tRNA genes, simple repeats and satellite regions. Internal similarity of each genome's TE complements was analyzed with *blastn* self-alignments of all TEs (considering a minimum 70% identity and 80 bp alignment length), and the distribution of percentage identity values was plotted using R.

## Analysis of gene microsynteny by ortholog pair collinearity

We used the frequency of collinear ortholog pairs as a proxy to estimate microsynteny across holozoans. Specifically, we retrieved all sets of single-copy orthologs for each pairwise species comparison within our set of 10 unicellular holozoan genomes. We then defined collinear gene pairs for each species pairs if the same two orthologs were adjacent in both genomes (irrespective of individual gene orientation to account for possible local inversions, as in (*Putnam et al., 2007*)). To account for spurious conservation of gene order, we assigned random positions to each gene using the bedtools v2.24.0 *shuffle* utility (*Quinlan and Hall, 2010*) in 100 independent rounds, for which the number of spurious conserved syntenic pairs was recorded. Then, we calculated the gene synteny ratio *r* of each species pair *i-j* as follows:

$$r_{ij} = \frac{\left(\dfrac{c_{ij} - s_{ij}}{N_{ij}}\right)}{\left(\dfrac{c_{max} - s_{max}}{N_{max}}\right)}$$

where *c* denotes the number of syntenic orthologs between *i* and *j*; *s* is the number of spurious syntenic orthologs averaged over 100 random replicates; and *N* is the number of comparable ortholog pairs between *i* and *j*. Values are normalized to the 0–1 interval using the maximum values of the

dataset as a reference, i.e. *Sphaeroforma* and *Creolimax*. A heatmap representing the degree of similarity in pairwise species comparisons was produced using the synteny ratio (R gplots library (*Warnes et al., 2016*)). Species were clustered according to their mean synteny. The same analysis was performed using the database of 40 eukaryotic genomes, which excluded *Abeoforma* and *Pirum*. In this case, the maximum values used as a reference were the *Nematostella-Aiptasia* pair.

For specific selected species comparisons, syntenic pairs were plotted onto the genome scaffolds using Circos v0.67 (*Krzywinski et al., 2009*).

## Analysis of coding sequence conservation

From our ortholog database using 40 eukaryotic genomes (excluding *Pirum* and *Abeoforma*, which had lower-quality gene annotations due to their fragmented assemblies), we selected 143 orthoclusters present in all unicellular holozoans, plus *Amphimedon queenslandica*, *Trichoplax adhaerens*, *Homo sapiens* and *Nematostella vectensis* (as representative animal genomes). We aligned each group of orthologs using MAFFT G-INS-i (*Katoh and Standley, 2013*), trimmed the alignments using trimAL automated algorithm (*Capella-Gutiérrez et al., 2009*), and inferred maximum likelihood trees for each ortholog group using RAxML v8.2.0 (*Stamatakis, 2014*) and the LG amino acid substitution model. Then, for each tree, we recorded all pairwise phylogenetic distances between species as measured by substitutions per alignment position using the cophenetic module of the ape v3.5 R library (*Paradis et al., 2004*; *Core Team, 2015*). We retrieved distances between each unicellular holozoan ortholog and, separately, *Amphimedon*, *Trichoplax*, *Homo* and *Nematostella* orthologs. For each inter-species comparison, we tested the significance of differences in phylogenetic distances between unicellular holozoans, using the non-parametric Wilcoxon rank sum test from the R stats library (*Core Team, 2015*).

## Comparative analysis of intron content

Intron content of a subset of 40 eukaryotic genomes (excluding *Abeoforma* and *Pirum*, which had lower-quality gene annotations due to their fragmented assemblies) was analyzed using a set of single-copy orthologous genes, and used to reconstruct ancestral states as described by Csűrös *et al.* (*Csuros et al., 2011*; *Csurös et al., 2007*, *Csurös et al., 2008*). We then selected orthocluster present as single-copies in 80% of our species dataset, allowing for paralog genes to occur in just one species per group (if that was the case, the best-scoring copy in BLAST alignments was kept). This yielded a group of 342 nearly paneukaryotic genes, whose protein translations were then aligned using MAFFT v7.245 G-INS-i algorithm (*Katoh and Standley, 2013*) and annotated with their intron coordinates (retrieved from their respective genome annotations). With this information, we reconstructed the ancestral states of each intron using the Malin implementation of the probabilistic model of intron evolution developed by Csűrös *et al.* (*Csűrös and Miklós, 2006*; *Csurös, 2008*), starting from the standard null model, running 1000 optimization rounds (likelihood convergence threshold = 0.001) and assuming a consensus eukaryotic phylogeny (see *Generation of a species tree for comparative analyses*).

Conserved intron sites (defined as unambiguously aligned in 80% of the orthologs, maximum of 10% of gap positions) were used to calculate the rates of intron loss and gain for each node of the tree. These rates were used to calculate a table of intron sites with a certain probability of presence, gain or loss at every node of the tree (which, when summed, give the number of introns that are present, gained or lost at that node (*Csűrös and Miklós, 2006*)). We computed 100 bootstrap replicates in Malin to assess uncertainty about inferred rate parameters and evolutionary history. In particular, we calculated the variance-to-mean ratio of the inferred number of introns in each ancestor with 100 bootstrap replicates (with values higher than one indicating more dispersed results and less reliable inferences).

For each node $i$, we calculate the percentage of introns gained ($p_{G,i}$)or lost ($p_{L,i}$) as a percentage of the total number of introns at that node. Then, the gain/loss ratio of a node, $r_i$, was calculated as follows:

$$p_{G,i} > p_{L,i} \rightarrow r_i = log_{10}\left(\frac{p_{G,i}}{p_{L,i}}\right) p_{L,i} < p_{L,i} \rightarrow r_i = log_{10}\left(\left(\frac{p_{G,i}}{p_{L,i}}\right)^{-1}\right) \times -1$$

We represented the presence and absence of intron sites at each lineage (extant and ancestral),

and the number of introns shared between species (only extant), using heatmaps (R gplots library (*Warnes et al., 2016*)). Inter-species distances were calculated using the pairwise counts of shared introns and the Spearman correlation algorithm, which was used to perform Ward hierarchical clustering as implemented in R stats library (*Core Team, 2015*). We used the same algorithms to calculate distances of intron presence probability profiles, and subsequent clustering.

For *Capsaspora*, the phylostratigraphy of intron sites was combined with the nucleosome-free sites identified by ATAC-seq analysis in (*Sebé-Pedrós et al., 2016b*), which were assumed to be putative regulatory sites. Then, we compared phylostrata distribution ('ancestral' *versus* 'recent' *Capsaspora*-specific sites) for introns with and without regulatory sites, using a Fisher's exact test: 74 recent introns and 465 ancestral introns lacked putative regulatory sites ($\geq$50% ATAC site overlap with the intron sequence, calculated using bedtools v2.24.0 *intersect* utility (*Quinlan and Hall, 2010*)), while 3 and 22 recent and ancestral introns had regulatory sites.

## Comparative analysis of protein domain architecture evolution

Protein domain architectures of the 40 eukaryotic species subset (excluding *Abeoforma* and *Pirum*, which had lower-quality gene annotations due to their fragmented assemblies) were computed using Pfamscan and the 29th release of the Pfam database (*Punta et al., 2012*). For each protein, the domain architecture was decomposed into all possible directed binary domain pairs (ignoring repeated consecutive domains; i.e. from protein A-B-B-C, the pairs A-B, A-C and B-C were built), and linked to its presence in its corresponding orthocluster (see *Generation of a species tree and ortholog datasets for comparative analyses* section). The final output was a numerical profile of species distribution for each combination of domain pairs in orthoclusters (considering that a cluster can contain more than one pair, and a pair can be present in more than one cluster, and thence the number of occurrences is recorded).

The numerical profile was analyzed using the general phylogenetic birth-and-death model developed by Csűrös and Miklós (*Csűrös and Miklós, 2006*) as implemented in Count (*Csurös, 2010*). This allows the comparative analysis and ancestral reconstruction of discretized quantitative properties of genomes, assuming a specific species tree (see *Comparative analysis of intron content*). We used a gain-loss-duplication model with unconstrained gain/loss and duplication/loss ratios in all lineages, assuming a Poisson distribution of orthocluster size at the LECA (root) and no rate variation categories. In this context, 'gain' was defined as the acquisition of a new pairwise domain combination in an orthocluster; a 'duplication' as the propagation of the combination (by gene duplication or convergent domain rearrangements); and 'loss' as pair dissociation. Starting from the standard null model, we ran 100 optimization rounds (convergence threshold = 0.1).

To analyze the modularity of the protein domain networks (and subnetworks) for each genome, we 1) calculated the community structure of each network using Louvain iterative clustering to obtain communities of domain pairs (undirected graphs), and 2) calculated the global network modularity according to these communities. The modularity parameter measures the fraction within-community edges minus the expected value obtained from a network with the same communities but random vertex connections (*Newman and Girvan, 2004*). A maximum value of 1 indicates a strong community structure, while a minimum value of 0 indicates that within-community edges are as frequent as expected in a random network. For these analyses we used the relevant algorithms from the igraph R library v1.0.1 (*Core Team, 2015*; *Csárdi and Nepusz, 2006*). Function-oriented domain subnetworks were obtained by retrieving orthologous groups that contained relevant domains, which were obtained from previous studies (transcription factors from (*de Mendoza et al., 2013*; *Weirauch and Hughes, 2011*), signaling domains from (*Richter and King, 2013*), ECM-related domains from (*Richter and King, 2013*; *Sebé-Pedrós et al., 2010*; *Hynes, 2012*), ubiquitination from (*Grau-Bové et al., 2015*)) and pfam2go annotations (for the subsets mentioned above, and also for protein-binding domains) (*Mitchell et al., 2015*). Monotonic statistical dependence between modularity and the number of domains per community was tested using Spearman's rank correlation coefficient ($\rho_s$) for all network or subnetwork (for original and simulated data).

## Comparative analysis of individual protein domain evolution

We mapped the presence of individual protein domains across our dataset of 40 eukaryotic species (excluding *Abeoforma* and *Pirum*), as predicted by Pfamscan and the 29th release of the Pfam

database (*Punta et al., 2012*). Using this numerical profile of domain presence in extant genomes, we computed the gains and losses at ancestral nodes using the Dollo parsimony algorithm as implemented in Count (*Csurös, 2010*).

## Accession numbers

Genome sequencing and assembly data from *Corallochytrium*, *Abeoforma*, *Pirum* and *Chromosphaera* has been deposited in NCBI using the BioProject accession PRJNA360047. Transcriptome sequencing data from *Abeoforma* and *Chromosphaera* has been deposited in NCBI using the BioProject accession PRJNA360056.

# Acknowledgements

This work was supported by an ERC Consolidator Grant (ERC-2012-Co-616960), support from the Secretary's Office for Universities and Research of the Generalitat de Catalunya (project 2014 SGR 619) and two grants from the Spanish Ministry for Economy and Competitiveness (MINECO; BFU2011-23434 and BFU2014-57779-P, the latter with European Regional Development Fund support), all to IRT. XGB was supported by a pre-doctoral FPI grant from MINECO (except for the January-March 2015 period). GT was funded by a European Marie Skłodowska-Curie Action (704566 AlgDates). HS was supported by JSPS KAKENHI 16K07468 and research grants from the NOVARTIS foundation for the Promotion of Science, ITOH Science Foundation, and JUTEN grant from the Prefectural University of Hiroshima. We thank Krista M. Nichols (NOAA Fisheries) and Chris Whipps (SUNY-ESF) for sharing their assembly of *Ichthyophonus hoferi*. We warmly acknowledge the help of Arnau Sebé-Pedrós and Alex de Mendoza for their invaluable comments on the manuscript; and thank Meritxell Antó, Elisabeth Hehenberger, Manuel Irimia, David López-Escardó, Jordi Paps, Dan Richter, and Valèria Romero-Soriano for their technical assistance and insightful remarks on the study. We thank Sebuastián R. Najle for his insightful observations on *Chromosphaera perkinsii*, which led to its naming.

# Additional information

## Funding

| Funder | Grant reference number | Author |
| --- | --- | --- |
| Ministerio de Economía y Competitividad | BFU2014-57779-P | Iñaki Ruiz-Trillo |
| European Commission | ERC-2012-Co -616960 | Iñaki Ruiz-Trillo |
| Ministerio de Economía y Competitividad | BFU-2011-23434 | Iñaki Ruiz-Trillo |
| Generalitat de Catalunya | 2014 SGR 619 | Iñaki Ruiz-Trillo |

The funders had no role in study design, data collection and interpretation, or the decision to submit the work for publication.

## Author contributions

XG-B, Conceptualization, Resources, Data curation, Formal analysis, Validation, Investigation, Methodology, Writing—original draft, Writing—review and editing; GT, Resources, Data curation, Formal analysis, Writing—review and editing; SD, Resources, Writing—review and editing; HS, GL, TAR, Resources, Data curation, Writing—review and editing; IR-T, Conceptualization, Supervision, Funding acquisition, Investigation, Project administration, Writing—review and editing

## Author ORCIDs

Xavier Grau-Bové, http://orcid.org/0000-0003-1978-5824
Guifré Torruella, http://orcid.org/0000-0002-6534-4758
Thomas A Richards, http://orcid.org/0000-0002-6096-750X
Iñaki Ruiz-Trillo, http://orcid.org/0000-0001-6547-5304

## Additional files

### Major datasets

The following datasets were generated:

| Author(s) | Year | Dataset title | Dataset URL | Database, license, and accessibility information |
|---|---|---|---|---|
| Xavier Grau-Bové, Meritxell Antó, Iñaki Ruiz-Trillo | 2017 | Genome sequencing and assembly data from Corallochytrium, Abeoforma, Pirum and Chromosphaera | https://www.ncbi.nlm.nih.gov/bioproject/PRJNA360047 | Publicly available at the NCBI BioProject database (accession no: PRJNA360047) |
| Xavier Grau-Bové, Meritxell Antó, Iñaki Ruiz-Trillo | 2017 | Transcriptome sequencing data from Abeoforma and Chromosphaera | https://www.ncbi.nlm.nih.gov/bioproject/PRJNA360056 | Publicly available at the NCBI BioProject database (accession no: PRJNA360056) |

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
