## [Decision Letter]

Thank you for submitting your article "Dynamics of genomic innovation in the unicellular ancestry of animals" for consideration by *eLife*. Your article has been reviewed by two peer reviewers, and the evaluation has been overseen by Diethard Tautz as the Senior Editor and Reviewing Editor.

The reviewers have discussed the reviews with one another and the Reviewing Editor has drafted this decision to help you prepare a revised submission.

Summary:

This is a useful, thorough, and well executed study to examine evolutionary origins of multicellularity in metazoa genome sampling of sister groups to the Metazoa (protists from the Holozoa clade – three Ichthyosporea and *Corallochytrium limacisporum*). The authors are inferring likely ancestral states of gene/genome characteristics that predated the emergence of multicellularity to highlight the changes that occurred with or enabled these traits. Some interpretation of how Metazoa complexity arose is possible with this dataset and the authors use these comparative analyses to provide some ideas about processes and consequences of these changes.

The *Corallochytrium* genome sequence is the most important contribution of this work to the data sets of eukaryotic genomes, as this lineage remained untouched by genomics as well as basically any other studies. Very little is known about this coral symbiont or parasite or commensal and more analysis of this genome in order to find out more about the biology of this organism would have been interesting, but the current paper focuses on the comparative analyses. This has been done very carefully and it is presented on 37 graphs and tables. The data are analysed very thoroughly and in sophisticated ways.

Essential revisions:

1) Given this huge amount of detailed information, it appears that the major take home message gets a bit lost. The Abstract suggests that "genome architecture evolution was equally dynamic [as gene evolution]: an early burst of gene diversity in the holozoan ancestor was followed by independent episodes of synteny disruption, intron gain, and genome expansions in both unicellular and multicellular lineages." But some discussion would be required whether the same picture would be seen in many other eukaryotic groups. A better focus on features in the genome evolution which could play a role in evolution of animal multicellularity would be warranted. If the authors do not see any, they should say it clearly as well because this is also a relevant finding.

2) The discussion of the intronization events – these can be interpreted as to how the change in gene structure and the timing of these changes relate, but this can also be interpreted as neutral processes due to changes in effective population size. Is there an adaptive interpretation which is hinted at, given the utility of introns as enabling alternative splicing? Please clarify.

3) The use of Regulated Unproductive Splicing and Translation through Nonsense Mediated Decay and "poison cassettes" (e.g. Lareau et al. Nature 2007, Lareau & Brenner MBE 2015) has been shown to have originated perhaps in the Opisthokont ancestor – is there any information about how the different intronization events could have been driven by differential NMD evolution? Are the splicing factors and NMD machinery present in these Holozoa?

4) Was there any signal in the protein domain network generation of and evidence of more promiscuous domains – in terms of enzymatic, structure proteins or any signals that relate to protein-protein interactions changes too? Or is this all fully captured in the graphs? One cannot tell if the types of domains that are playing a role in the changes (seen increasing in 6B – Metazoa) are mostly the classes highlighted at right or if any others come into play – if the authors are taking a candidate family focused approach here but are also able to discover other types of expansions?

5) The last paragraph is unclear:

"The genomes of extant Metazoa are subject to overlapping evolutionary dynamics for different traits" – what makes any of this pre or post-metazoan ancestor specific? All genomes change and why wouldn't we expect these dynamics to be a constant refrain back to the unicellular ancestor?

6) Similarly, the final sentence summarizes the interpretations: "Consequently, we see how the genomes of ancestral premetazoans were subject to the same processes observed in most animal phyla" – it is unclear why we should expect anything different. What aspects of the animal phyla have driven the different processes? effective population size changes? Expansion of cell type or true multicellularity?

7) Last sentence of Discussion, last phrase:

"and no trait left to tinker with" – this implication is difficult to understand. Why is there nothing left to tinker with?

---

## [Author Response]

Essential revisions:

1) Given this huge amount of detailed information, it appears that the major take home message gets a bit lost. The Abstract suggests that "genome architecture evolution was equally dynamic [as gene evolution]: an early burst of gene diversity in the holozoan ancestor was followed by independent episodes of synteny disruption, intron gain, and genome expansions in both unicellular and multicellular lineages." But some discussion would be required whether the same picture would be seen in many other eukaryotic groups. A better focus on features in the genome evolution which could play a role in evolution of animal multicellularity would be warranted. If the authors do not see any, they should say it clearly as well because this is also a relevant finding.

We appreciate the concern regarding the clarity of the message. We have introduced some changes that we believe will help clarify it. First, in order to simplify the Abstract, we now refer exclusively to the “episodes of synteny disruption, intron gain, genome expansions” that occurred, specifically, in the immediate ancestry of Metazoa – which are of interest to a wider readership interested in multicellularity origins. Other reorganizations of the genome occurring in unicellular holozoan groups, which also feature prominently in our results, are then discussed on a per-analysis basis.

We have also now more clearly introduced the relationship between ‘genomic complexity’ and multicellularity earlier on in the Introduction section (second paragraph). In the pertinent sections of the Results and Discussion, we further elaborate on some of these issues. For example, we explore the link between intron gains and richer alternative splicing profiles (via isoform productions or NMD) with new analyses (following comments #2 and #3; see below for details).

Finally, the re-writing of the concluding paragraph of the article (in accordance with revisions #5, #6 and #7) is also meant to enhance the article’s take-home message. We indeed now mention the apparent lack of correlation between increasing organismal and genomic complexity in Holozoa.

2) The discussion of the intronization events – these can be interpreted as to how the change in gene structure and the timing of these changes relate, but this can also be interpreted as neutral processes due to changes in effective population size. Is there an adaptive interpretation which is hinted at, given the utility of introns as enabling alternative splicing? Please clarify.

We understand the reviewer’s point and we have updated our manuscript to cover these issues. Although the original manuscript mentioned the potential adaptive roles at the beginning of the corresponding Results sub-section (“Intron evolution in Holozoa: two independent 'great intronization events”), this was a rationale for the analyses rather than a thorough discussion. In order to amend this shortcoming and to cover the possibility of neutral processes, we have expanded both the Results and Discussion sections at the following points:

Results subsection “Consequences of intron gains in early holozoan evolution”: we now debate the evolutionary implications of intron gain episodes in Holozoa, first mentioning the possibility of a drift-driven change, and then elaborating on the nature of alternative splicing in non-animal eukaryotes (second paragraph). Since unicellular holozoans’ AS was likely dominated by intron retention (as seen in most eukaryotes (Irimia and Roy 2014; McGuire et al. 2008), *Capsaspora* (Sebé-Pedrós et al. 2013) and *Creolimax* (de Mendoza et al. 2015)), we link this argumentation with the survey of NMD and splicing factors’ evolution suggested by the reviewers (last paragraph; please see the answer to essential revision #3 for further details).

Discussion subsection “Dating the origin of animal-like protein domain architectures, intron architecture and genome size”, second paragraph: we have added a more in-depth discussion of the evolutionary implications of intron gains in ancestral Holozoa (both in Metazoa and ichthyosporeans). We now cover the relative importance of exon skipping and intron retention in unicellular and multicellular holozoans (already present in the original submission); the fact that being able to perform NMD could have facilitated the intron invasion according to the population-genetic hypothesis mentioned in the reviewer’s suggestion; and the adaptive potential associated to high transcriptome variability in intron-rich genomes.

In the Discussion section (aforementioned paragraph) we have added an assessment of the relative importance of adaptive and non-adaptive/neutral changes in intron evolution. Essentially, our data forces us to remain agnostic: population-level data is generally unavailable for unicellular holozoans (see forthcoming paragraph), and in-depth transcriptomic analyses are required to demonstrate specific adaptations by AS. We thus conclude that both mechanisms could have been in place, but that their relative importance remains an open debate.

Finally, we would like to expand on the lack of data concerning the effective population size of unicellular holozoans. This shortcoming has limited our ability to explicitly test the neutral hypothesis mentioned by the reviewers. There are two reasons for which we do not have confident estimations of the effective population size of our unicellular holozoans:

Multiple isolates are generally unavailable: the species here analysed are scarce in environmental surveys, and thus relatively difficult to isolate more than once (de Vargas et al. 2015). As a reference, the 7 teretosporeans included in this project are represented by ~30,000 reads of the 18S gene in the whole TARA Oceans global survey (99% of them correspond to *Abeoforma*, the others being virtually absent). These abundances are low: they represent just 0.006% of the eukaryotic reads, or 0.013% of the opisthokonts’. The only species for which population-level data is available (already mentioned in the manuscript; subsection “Consequences of intron gains in early holozoan evolution”, first paragraph) is *Sphaeroforma tapetis*, for which a typical eukaryotic effective population was inferred (10^[104]^-10^[139]^) from multiple natural isolates (Marshall and Berbee 2010).

The available cultures were grown in the lab for many years before sequencing, which precludes the indirect estimation of population sizes from heterozygosity levels due to the population bottlenecks imposed by isolation and artificial culture pressures.

3) The use of Regulated Unproductive Splicing and Translation through Nonsense Mediated Decay and "poison cassettes" (e.g. Lareau et al. Nature 2007, Lareau & Brenner MBE 2015) has been shown to have originated perhaps in the Opisthokont ancestor – is there any information about how the different intronization events could have been driven by differential NMD evolution? Are the splicing factors and NMD machinery present in these Holozoa?

We agree this is a very interesting question. We have incorporated this suggestion, which we see as an interesting improvement of the manuscript that has enhanced our discussion of intron evolution regarding the major comment #2 as well (above).

Thus, we have performed an evolutionary survey of the core NMD machinery, as defined in a recent review (He and Jacobson 2015), as well as the wider set of SR splicing factors (including SRSF5, mentioned by the reviewer) that are known to be involved in alternative splicing. This analysis is presented jointly with the intron site evolution analyses, and is shown in Figure 5, Figure 5—figure supplement 2 and 3 and [Supplementary-material SD7-data].

Specifically, we have found that:

The NMD core machinery is ancestral to all eukaryotes and is thoroughly conserved in all the holozoan LCAs that underwent intron expansions (Holozoa, Ichthyophonida, Metazoa, Choanoflagellata…). However, it has undergone a notable reduction in *Corallochytrium,* concordantly with its reduced intron content.

The animal SR splicing factors (according to their nomenclature in humans: SRSF1-9 andTRA2A-B) are present in all unicellular ancestors of Metazoa but have undergone partial secondary reductions in some unicellular holozoan lineages. As with NMD, *Corallochytrium* has the most depleted SR complement among unicellular holozoans.

Other RNA-binding motifs that can intervene in the splicing process are nevertheless common in unicellular holozoans, with ~100 motifs per genome – in line with other eukaryotes, but less than most metazoans (~300 in *H. sapiens*, ~200 in *N. vectensis*, etc.).

We thus conclude that the intron gain episodes of holozoans occurred in ancestral organisms that were potentially able to perform NMD of aberrant transcripts. This result fits the proposal that intron gain episodes can be linked with active NMD, as regulated removal of aberrant transcripts can decrease the cost of increased intron contents (Lynch 2006). These results are now exposed in the Results section (subsection “Consequences of intron gains in early holozoan evolution”) and further explored in the Discussion (subsection “Dating the origin of animal-like protein domain architectures, intron architecture and genome size”; please see response to comment #2 for further details). We have also updated the Materials and methods section to explain the homolog search procedure used for each of these gene families (subsection “Retrieval of homologous sequences”).

4) Was there any signal in the protein domain network generation of and evidence of more promiscuous domains – in terms of enzymatic, structure proteins or any signals that relate to protein-protein interactions changes too? Or is this all fully captured in the graphs?

Thanks for bringing up this point. Although this was not thoroughly discussed in the main text, we did have analysed the promiscuities of domains involved in protein-protein interactions (category “protein binding” in Figure 7). We find that the sub-set of protein-binding domains generally diversify at higher rates than the rest of the proteome in the animal ancestry and also in the direct LCAs of extant unicellular holozoans (green shades in Figure 7’s rightmost panel). We agree that, given that protein-protein interaction domains are often among the most promiscuous motifs according to several studies, exploring these results helps putting the domain promiscuity analysis in context. We now explore these results as follows:

“This same effect was observed when we analyzed subsets of the proteome networks sharing a common function: the diversification of gene families with domains related to the ECM, signaling, ubiquitination or protein-protein interactions occurs by acquisition of promiscuous domains that reduce their modularity (with ρs in the range -0.32 to -0.84 and p<0.001; Figure 8—figure supplement 1), and this reduction is frequently stronger in animals than in their unicellular relatives and ancestors (Figure 8—figure supplement 1). The high promiscuity of domains mediating protein-protein interactions has already been reported in previous analyses [Marshall and Berbee, 2010; Tordai et al. 2005], thus confirming the validity our approach.”

Furthermore, in order to make clear the contribution of protein-binding domains to the higher domain promiscuities exhibited by metazoans, we have expanded the Figure 8 supplement to include the per-species promiscuities of the sub-networks mentioned in Figure 7’s right panel: ECM, signalling, ubiquitination and protein-binding domains, in addition to TFs (already shown in the main Figure 8).

One cannot tell if the types of domains that are playing a role in the changes (seen increasing in 6B – Metazoa) are mostly the classes highlighted at right or if any others come into play – if the authors are taking a candidate family focused approach here but are also able to discover other types of expansions?

As the reviewers’ note, we took a candidate family-focused approach using pre-defined sets of protein domains that were manually curated in previously published analyses (subsection “Comparative analysis of protein domain architecture evolution”, last paragraph; Methods section). Readers that seek to examine the complete lists of protein domain pairs gained in a specific ancestor can refer to the data-set made available with our study ([Supplementary-material SD11-data], in the form of a readable spreadsheet). To clarify this possibility, we now mention it in the Results:

“Our ancestral reconstruction of protein domain architectures ([Supplementary-material SD11-data]) allowed us to investigate the evolutionary origin of specific domain organizations within gene families and examine their diversification pattern in the ancestry of animals (Table 1—source data 1).”

In order to facilitate the interpretation of the data presented in [Supplementary-material SD11-data] (gains of protein domain combinations in various ancestors), we have added a new source data table (Table 1—source data 1) that includes all the protein domain combinations present in the LCA of Metazoa and their gain probabilities in eukaryotic ancestral nodes. Table 1—source data 1 has the same format as Table 1 and is directly produced from the [Supplementary-material SD11-data] dataset; thus, readers can readily build similar tables for their nodes or gene families of interest.

Finally, we would like to argue in favour of our candidate-focused approach. Automatic associations of protein domains and functional terms (e.g., gene ontologies [GO]) in unicellular holozoans are heavily biased by studies in animal model organisms. For example, protein domains that appear in the animal ancestry (TFs, ECM…) are often associated with GO terms that are meaningless in unicellular contexts (development, organismal communication…). This dissonance obscures the interpretation of downstream analyses such as GO enrichment tests, as it becomes difficult to distinguish functional mis-annotations from real but imprecisely-named GOs. Relying on curated lists of domains helps overcome this problem.

5) The last paragraph is unclear:

"The genomes of extant Metazoa are subject to overlapping evolutionary dynamics for different traits" – what makes any of this pre or post-metazoan ancestor specific? All genomes change and why wouldn't we expect these dynamics to be a constant refrain back to the unicellular ancestor?

6) Similarly, the final sentence summarizes the interpretations: "Consequently, we see how the genomes of ancestral premetazoans were subject to the same processes observed in most animal phyla" – it is unclear why we should expect anything different. What aspects of the animal phyla have driven the different processes? effective population size changes? Expansion of cell type or true multicellularity?

We took the liberty of answering remarks #5 and #6 together, as they are both related to similar statements in the last paragraphs of the manuscript. We agree that the above-mentioned sentences convey a sense of opposition between genome evolution pre- and post-metazoan origins. We would not agree with this proposition as a general principle, neither under the light of our own results nor of the accumulating knowledge regarding genome evolution (as the reviewers highlight). Furthermore, previous investigations of holozoan genomics regarding gene content actually led us to expect quite the opposite: that unicellular relatives of animals, extant or ancestral, do not have “simpler” genomes with “static” architectures/contents. Indeed, we formalized this expectation in our study whenever possible – e.g., we hypothesized that intron-richer ichthyosporeans and animals would descend from a single pan-holozoan intron gain episode (admittedly, the hypothesis was false).

We have completely rewritten this paragraph to amend these incongruences (subsection “Genomic innovation in the animal ancestry”). We now start by overviewing the classical features of complex animal genomes, some of which are also frequent (and actually emerged) in premetazoan lineages. Then, we discuss how our analyses have clarified the timeline of genome architecture evolution (relatively overlooked outside of metazoans) and summarize the overlapping evolutionary dynamics described in the paper. In this concluding paragraph, we nevertheless refrain from linking our results with changes of effective population sizes/cell type origin/true multicellularity: as explained above, we prefer to refer to these issues on a per-analysis basis, earlier on in the Discussion section.

Incidentally, we think that amending this last paragraph also helps to address the concerns expressed in the comment #1 regarding the unclear take-home message.

7) Last sentence of Discussion, last phrase:

"and no trait left to tinker with" – this implication is difficult to understand. Why is there nothing left to tinker with?

This particular sentence was meant to emphasize, in a literary manner, the dynamism of genome evolution in unicellular animal ancestors. It has similar implications as the controversial sentences indicated in comments #5 and #6. Thus, as part of the rewriting of this paragraph (subsection “Genomic innovation in the animal ancestry”), we have now deleted this sentence.